# TRANSFERRING PRETRAINED DIFFUSION PROBABILISTIC MODELS

## ABSTRACT

Diffusion Probabilistic Models (DPMs) achieve impressive performance in visual generative tasks recently. However, the success of DPMs heavily relies on large amounts of data and optimization steps, which limits the application of DPMs to small datasets and limited computational resources. In this paper, we investigate transfer learning in DPMs to leverage the DPMs pretrained on large-scale datasets for generation with limited data. Firstly, we show that previous methods like training from scratch or determining the transferable parts is not suitable for the DPM due to its U-Net based denoising architecture with the external denoising timestep input. To address it, we present a condition-based tuning approach to take full advantages of existing pretrained models. Concretely, we obtain the semantic embeddings of condition images by the pretrained CLIP model, and then inject these semantic informations to the pretrained DPM via a "Attention-NonLinear" (ANL) module. The adaptation to a new task can be achieved by only tuning the ANL module inserted into the pretrained DPM hierarchically. To further enhance the diversity of generated images, we introduce a masked sampling strategy based on the condition mechanism. Extensive experiments validate the effectiveness and efficiency of our proposed tuning approach in generative task transfer and data augmentation for semi-supervised learning.

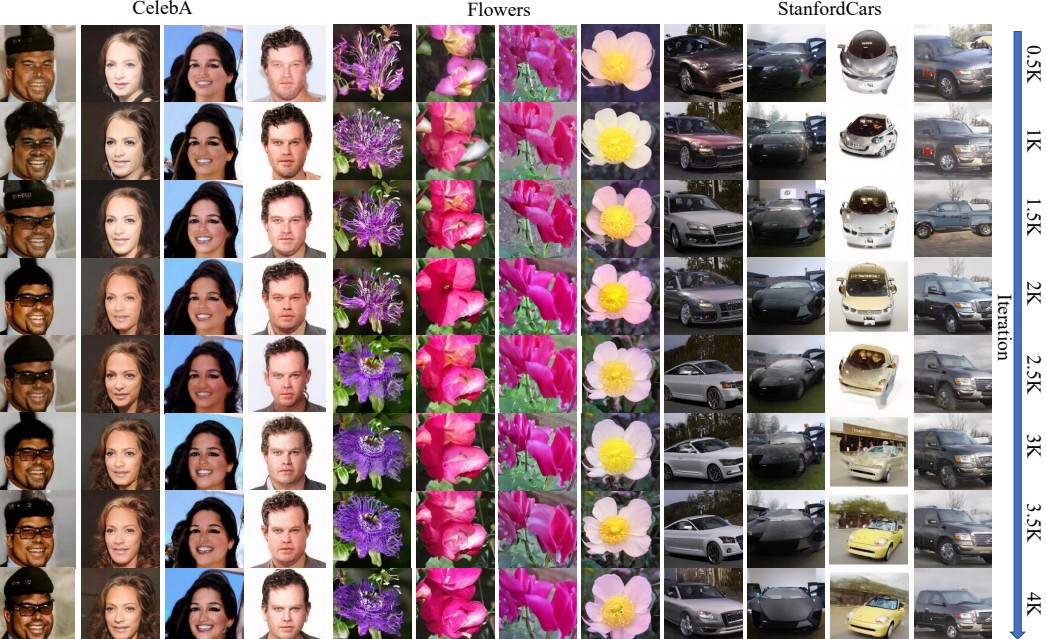

Figure 1: Generated images by tuning the latent diffusion model Rombach et al. (2022) pretrained on ImageNet Krizhevsky et al. (2012) with our proposed tuning approach (batch size=32). We illustrate the generated images from iteration 500 to 4000 with the interval of 500 on three datasets: CelebA Liu et al. (2015), Flowers Nilsback & Zisserman (2008) and StanfordCars Krause et al. (2013).

# 1 INTRODUCTION

Recently, diffusion probabilistic models (DPMs) (Ho et al., 2020; Song et al., 2020b) have been demonstrated the increasing power of generating complex and high-quality images by introducing a hierarchy of denoising autoencoders. DPMs adopt a diffusion process to gradually inject noise to the data distribution, and then learn the reverse process to generate images by a Markov model. However, the success of diffusion models heavily depends on huge data and computation costs. For instance, training DPMs often needs large-scale datasets (e.g., ImageNet (Krizhevsky et al., 2012)), and takes hundreds of GPU days in extreme cases (e.g., 150-1000 V100 GPU days in (Dhariwal & Nichol, 2021)). Considering that it is usually hard to collect adequate training data for a specific domain (Long et al., 2015), we propose to leverage the released powerful DPMs pretrained on large-scale datasets to facilitate downstream generative tasks.

This transfer learning paradigm is thoroughly investigated on discriminative tasks in many fields like computer vision (Bengio, 2012; Yosinski et al., 2014; Zamir et al., 2018) and natural language processing (Devlin et al., 2018; Mozafari et al., 2019; Peng et al., 2019). For computer vision, Yosinski et al. (2014) observed that the representations learned by deep convolutional neural networks transit from general (e.g., Gabor filters and color blobs) to specific, which facilitates the "pretraining and fine-tuning" learning paradigm. For natural language processing, reusing the pretrained BERT model achieves impressive performance on multiple downstream tasks (Houlsby et al., 2019).

However, few efforts has been paid to investigate transfer learning in the generative tasks, especially for the DPMs. (Zhao et al., 2020) is a recent work, which explores the transferability of generative adversarial networks (GANs). Motivated by the insights from (Yosinski et al., 2014), they proposed to preserve the low-level layers that capturing properties of generic patterns, and fine-tune the high-level layers that are associated with the semantic aspects of data. However, the insight of determining the transferable parts is not suitable for DPMs since DPMs often adopt a U-Net based denoising architecture with the external denoising timestep input. DPMs generate images by progressively predicting the statistics of the gaussian noise corresponding to a specific timestep, so it is hard to say which part learns the general or task-specific representations.

We have also considered other two baselines: training from scratch and tuning the whole pretrained DPM ("Tuning-All"). Based on empirical observations, we find training from scratch is infeasible for learning with limited data and optimization steps, and Tuning-All is likely to overfit small training sets, resulting in slow convergence.

In this work, we present a condition-based tuning approach to achieve fast adaptation from pretrained DPMs to new datasets. To make the adaptation procedure efficient, we propose to take maximum advantages of pretrained models. Firstly, we leverage the vision-text pretrained model CLIP (Radford et al., 2021) to obtain the embedding of input images. It is worth noticing that the CLIP embedding is highly associated with the semantic informations of images. To make use of CLIP embeddings, we consider an "Attention-NonLinear" module (ANL), which injects the external conditions (i.e., the CLIP embedding) to each block of the pretrained DPM. The ANL module consists of a cross-attention and a non-linear mapping module. Note that the ANL module is inserted into each block of DPM, fusing the CLIP embedding and adjusting the DPM hierarchically. Therefore, transferring pretrained DPM to a new dataset can be achieved by fine-tuning the ANL modules only and freezing the pretrained CLIP model and DPM.

Introducing the trainable ANL modules into pretrained DPMs enjoys the following advantages: (1) This tuning approach preserves the main structures of pretrained DPMs, but is able to achieve effective transfer since the ANL module hierarchically adjusts each block of the pretrained DPM. (2) Compared to training from scratch and tuning the whole DPM, our approach introduces less trainable parameters and faster adaptation, which is beneficial for learning with limited data and training resources. (3) Based on the CLIP semantic embeddings, our approach can be extended to the language modality like simple text-to-image synthesis.

In summary, we have the following contributions:

1. We investigate transfer learning in recent DPMs, and uncover that previous methods like training from scratch or determining the transferable parts are not efficient. To the best of our knowledge, we are at the very early attempt to leverage pretrained DPMs for generation with limited data and training resources.

2. We present a condition-based tuning approach, that injects the CLIP embeddings to the pretrained DPM via an "Attention-NonLinear" (ANL) module. Our tuning approach has less parameters needed to be tuned and converges faster than the baselines.

3. We experimentally show that our tuning approach achieves fast adaptation with few training data and optimization steps, and outperforms previous generative transfer learning approaches. Further, we also evaluate the potential of our method in data augmentation for the data-scarce scenarios like semi-supervised learning.

## 2 BACKGROUND

**Diffusion Probabilistic Models (DPMs).** DPMs construct a forward process that gradually adds noise a data distribution $p(\mathbf{x}_0)$ and learn to reverse this process for generation. Based on the Markov forward process with a forward noise schedule $\beta_t \in (0,1), t = 1, \cdots, T$, denoising diffusion probabilistic models (DDPMs) formulate it as:

$$q(\mathbf{x}_{1:T}|x_0) = \prod_{t=1}^{T} q(\mathbf{x}_t|x_{t-1}), \qquad q(\mathbf{x}_t|x_{t-1}) = \mathcal{N}(\mathbf{x}_t; \sqrt{1-\beta_t}\mathbf{x}_{t-1}, \beta_t\mathbf{I}), \qquad (1)$$

where $\mathbf{I}$ is the identity matrix. The reverse (generation) process is defined as

$$p_\theta(\mathbf{x}_{0:T}) = p(\mathbf{x}_T) \prod_{t=1}^{T} p_\theta(\mathbf{x}_{t-1}|x_t), \qquad p_\theta(\mathbf{x}_{t-1}|\mathbf{x}_t) = \mathcal{N}(\mathbf{x}_{t-1}|\mu_\theta(\mathbf{x}_t, t), \Sigma_\theta(\mathbf{x}_t, t)), \quad (2)$$

where $\Sigma_\theta(\mathbf{x}_t, t)$ is often fixed as a constant. The reverse (generation) process can be learned by optimizing a variational bound on negative log-likelihood:

$$L = \mathbb{E}_q \left[ D_{\mathrm{KL}}(q(\mathbf{x}_T|\mathbf{x}_0)||p(\mathbf{x}_T)) + \sum_{t>1} D_{\mathrm{KL}}(q(\mathbf{x}_{t-1}|\mathbf{x}_t, \mathbf{x}_0)||p_\theta(\mathbf{x}_{t-1}|\mathbf{x}_t)) - \log p_\theta(\mathbf{x}_0|\mathbf{x}_1) \right]. \tag{3}$$

Let $\alpha_t = 1 - \beta_t$ and $\overline{\alpha}_t = \prod_{s=1}^{t}$, $\mathbf{x}(\mathbf{x}_0, \epsilon)$ can be reparameterized as $\sqrt{\overline{\alpha}_t}\mathbf{x}_0 + \sqrt{1-\overline{\alpha}_t}\epsilon$ for $\epsilon \in \mathcal{N}(0, \mathbf{I})$. By parameterizing $\mu_\theta$ with the predicted noise $\epsilon_\theta$, we have

$$L_{\mathrm{DDPM}} = \mathbb{E}_{\mathbf{x}, \epsilon \sim \mathcal{N}(0,\mathbf{I}), t} \left[ ||\epsilon - \epsilon_\theta(\sqrt{\overline{\alpha}_t}\mathbf{x}_0 + \sqrt{1-\overline{\alpha}_t}\epsilon, t)||^2 \right]. \tag{4}$$

Recently, Rombach et al. (2022) proposed to perform diffusion models on the latent code of images rather than the original ones, and named it as latent diffusion models (LDMs). Specifically, LDMs firstly train an autoencoder to convert the original images to a low-dimensional latent space that is perceptually equivalent to the original data space. Let $\mathcal{E}$ be the encoder that encodes $\mathbf{x}$ to a latent code $z$, $D$ be the decoder that reconstructs $\hat{\mathbf{x}}$ from the latent code $z$: $\hat{\mathbf{x}} = D(\mathcal{E}(\mathbf{x}))$, we can write the learning objective of LDMs

$$L_{\mathrm{LDM}} = \mathbb{E}_{\mathcal{E}(\mathbf{x}), \epsilon \sim \mathcal{N}(0,\mathbf{I}), t} \left[ ||\epsilon - \epsilon_\theta(z_t, t)||^2 \right]. \tag{5}$$

Since LDMs achieve state-of-the-art performance on image synthesis and are capable for high-resolution synthesis with less inference costs, we build our method upon the LDM pretrained on the large-scale dataset ImageNet (Krizhevsky et al., 2017) with the resolution of $256 \times 256$. To evaluate the performance of generative models, we use the Frechet Inception Distance (FID) metric (Heusel et al., 2017), which estimating the realism and variation of generated images.

## 3 METHOD

In this section, we elaborate how to train a DPM with limited data. Considering that training a DPM from scratch usually needs large amounts of data and training resources (Rombach et al., 2022), we propose to transfer a DPM pretrained on large-scale datasets to a new task with limited samples from distinct categories and domains. Specifically, given a pretrained DPM, we preserve the main structures, and inject external semantic conditions via an "Attention-NonLinear" (ANL) module. Hence, via this method, the number of parameters that need to be tuned is reduced, alleviating the overfit issue when learning with limited data. Here, we introduce our tuning approach in detail:

Figure 2: (Best viewed in color.) An overview of three approaches for task transfer with the U-Net based diffusion probabilistic model (DPM): (a) random initializing the DPM and learning from scratch, (b) tuning the whole DPM pretrained on large-scale datasets, (c) the proposed tuning approach, which takes advantages of pretrained models and only adapts the "Attention-NonLinear" module to achieve fast and effective transfer. Taking the latent diffusion model pretrained on ImageNet with the resolution of $256\times256$ as an example: the number of trainable parameters is 400M for (a) and (b), 169M for (c) our tuning approach.

## 3.1 CONDITION DPMs WITH CLIP EMBEDDING

Many approaches have been proposed for conditional DPMs like conditioning with a classifier (Nichol & Dhariwal, 2021), conditioning during sampling with a reference image (Choi et al., 2021), learning a conditional score (Song et al., 2020b) and injecting external conditions during learning (Rombach et al., 2022; Giannone et al., 2022). The condition mechanism is usually recognized as a technique for conditional generation like class-conditional generation and text-to-image synthesis. Recently, (Giannone et al., 2022) reveal that conditioning a DPM with rich and expressive information promotes learning. To achieve this, Giannone et al. (2022) introduce an additional trainable vision transformer for learning the conditional representations. However, both training the vision transformer and DPM may suffer from overfitting when limited target data are provided. Following the criteria that take full advantages of pretrained models, we propose to directly obtain the conditional representations by the vision-language pretrained model CLIP (Radford et al., 2021):

$$c = \phi(\mathbf{x}), \tag{6}$$

where $\phi$ is a pretrained CLIP model, $c$ is the semantic embedding corresponding to the image input $\mathbf{x}$. We take the LDMs as an example, the learning objective can be written as:

$$L := \mathbb{E}_{\mathcal{E}(\mathbf{x}), \epsilon \sim \mathcal{N}(0, \mathbf{I}), t} \left[ ||\epsilon - \epsilon_\theta(z_t, t, c)||^2 \right], \qquad c = \phi(\mathbf{x}). \tag{7}$$

CLIP is a powerful pretrained model that learning directly from 400 million text-image pairs. Compared to (Giannone et al., 2022), our approach shows the following advantages: (1) Using the CLIP embedding fastens the learning process since we get rid of training an additional embedding network. (2) Thanks to the shared text-image embedding space, we can achieve simple text-to-image generation even without training with explicit text-image pairs.

## 3.2 "ATTENTION-NONLINEAR" MODULES FOR EFFECTIVE TUNING

Given the semantic embedding $c$, we present an "Attention-NonLinear" (ANL) module that injects the rich semantic information to the pretrained DPM. The ANL module consists a cross-attention and a non-linear mapping module. In practice, DPMs usually adopt the UNet backbone, in which residual convolutional blocks are stacked hierarchically. In order to adapt each block of pretrained UNet, we add the ANL module after each residual convolutional block. Based on this, we present an efficient and effective tuning approach for transferring pretrained DPMs: tuning the ANL modules only, and freezing the pretrained residual convolutional blocks. Here, we elaborate each component of the ANL module by taking the pretrained LDM as an example.

The cross-attention module is used to fuse the intermediate representations of the pretrained LDM and the semantic embeddings of input images:

$$\text{Attention}(Q, K, V) = \text{softmax}(\frac{QK^T}{\sqrt{d_k}})V, \quad Q = W_Q^{(i)} \cdot h_i(z_t), \quad K = W_K^{(i)} \cdot c, \quad V = W_V^{(i)} \cdot c, \tag{8}$$

where $h_i(z_t) \in \mathbb{R}^{N, d^i}$ is the intermediate representation of $i$-th residual convolutional block, $c \in \mathbb{R}^{N, d_c}$ is the CLIP embedding of input $\mathbf{x}$ and independent of the timestep. $W_Q^{(i)} \in \mathbb{R}^{d_k, d^i}, W_K^{(i)} \in \mathbb{R}^{d_k, d_c}$ and $W_V^{(i)} \in \mathbb{R}^{d_k, d_c}$ are the learnable mapping matrices (Vaswani et al., 2017). Recalling that

we hope to preserve the main structures of pretrained DPMs, and adapt them with tuning a specific module, we propose a non-linear adapter that mapping the semantic-fused representations to match the pretrained LDM. Therefore, the ANL module can be written as:

$$\text{ANL}(h_i(z_t), c) = f(\text{Attention}(h_i(z_t), c)), \tag{9}$$

where $\text{ANL}()$ is the ANL module parameterized by $\theta_{\text{ANL}}$, $f$ is the non-linear mapping module consisting two linear layers and an activation layer, $\text{Attention}()$ is the cross-attention module defined in Eq. (8), fusing the intermediate representation of the pretrained LDM $h_i(z_t)$ and the semantic CLIP embedding of the image input $c = \phi(\mathbf{x})$. Here, we present our learning objective:

$$\min_{\theta_{\text{ANL}}} \mathbb{E}_{\mathcal{E}(\mathbf{x}), \epsilon \sim \mathcal{N}(0, \mathbf{I}), t} \left[ ||\epsilon - \epsilon_\theta(z_t, t, c)||^2 \right]. \tag{10}$$

To better illustrate the insight of our tuning approach, we discuss three baselines here: (i) training from scratch, (ii) tuning the whole pretrained DPM, and (iii) determining the transferable part of the pretrained DPM. Training a DPM from scratch usually needs large amounts of training samples and takes a long iterative process to converge (Nichol & Dhariwal, 2021). Training a DPM from scratch usually needs large amounts of data, and fails to generate meaningful images with limited training samples and optimization steps (Sec. 4.2). Tuning the whole model is able to adapt to a new task, but still introduces lots of parameters to be trained. The third method takes from (Zhao et al., 2020), which freezes the transferable part that extracts the general features and fine-tunes the remaining part that mainly learns the task-specific features to achieve task transfer. However, this paradigm is infeasible for DPMs. Because the timestep is an external input for DPM, and the output of DPM is the predicted mean and variance of gaussian noise. Hence, it is not reasonable to indicate which part is general or task-specific. An overview of baselines (i), (ii) and our approach is illustrated in Fig. 2.

### 3.3 Masked Sampling and Unconditional Sampling

In the sampling stage, our model takes an image as the condition input for conditional generation. Since the generation process is guided by the CLIP embedding, the generated images may be over-constrained by the training data. To ensure the diversity of generated images, we present a simple masked sampling strategy to mask part of the condition image. Concretely, given a mask rate $\delta \in (0, 1)$ and condition images $X \in \mathbb{R}^{bs, ch, he, wi}$ where $bs$ is the batch size, $ch$ is the number of channels, $he$ and $wi$ are the height and width, respectively, we generate the mask matrix $M \in \mathbb{R}^{he, wi}$ by masking a matrix with all values of 1. It can be easily implemented by identifying the coordinates of upper left corner $(\text{rand}(0, (1 - \delta)he), \text{rand}(0, (1 - \delta)wi))$ and lower right corner $(\delta he + \text{rand}(0, (1 - \delta)he), \delta wi + \text{rand}(0, (1 - \delta)wi))$, where $\text{rand}()$ is an integer random function. Then, we can obtain the masked images $X'$ by repeating the mask matrix to match the size of inputs. By forwarding the masked images to the CLIP model, we obtain the perturbed semantic representations for diverse conditional sampling.

Another extension of our work is unconditional generation. Recalling that the key insight of our work is the efficient tuning approach based on the ANL module, which can be generalized to the unconditional generative models. Specifically, we replace the cross-attention module with the self-attention module since unconditional generative models take no condition input. In other words, we tune the self-attention layers and non-linear mapping layers to achieve task transfer. By this way, the semantic encoder (e.g., CLIP) is discarded. Compared to the aforementioned conditional models, unconditional models take more optimization steps, but show more flexibility since they have no need of the condition input during sampling.

## 4 Experiments

In this section, we evaluate the proposed techniques in natural image generation and its potential application on semi-supervised learning. Specifically, following the protocols in (Zhao et al., 2020), we transfer the latent diffusion model (LDM) pretrained on the large-scale dataset ImageNet (Krizhevsky et al., 2012) to three smaller datasets: CelebA (Liu et al., 2015) (202,599), Flowers (Nilsback & Zisserman, 2008) (8,189), StanfordCars (Krause et al., 2013) (7,350) and their variants: CelebA-1k, Flowers-1k, StandfordCars-1k and the extreme scenario with 25 training samples. (Zhao et al., 2020) also conducts experiments on the Cathedral dataset, which is not released now. Further, we take our approach as a data augmentation technique, and demonstrate its effectiveness in semi-supervised learning (StandfordCars (Krause et al., 2013)).

Table 1: FID scores (↓) of GAN-based and DPM-based approaches after 60k iterations. "failed" indicates training or mode collapse. "*" means the results are obtained on CelebA-10k rather than CelebA-full.

| Method/Target | CelebA | Flowers | Cars | CelebA-1k | Flowers-1k | Cars-1k |
|---|---|---|---|---|---|---|
| TransferGAN | 18.69 | failed | failed | - | - | - |
| ScratchGAN | 16.51 | 29.65 | 11.77 | 20.75 | 58.18 | 39.97 |
| (Zhao et al., 2020) | 9.90 | 16.76 | **10.10** | 19.77 | 43.05 | 35.88 |
| ScratchDiff | *46.97 | failed | failed | failed | failed | failed |
| Tuning-All | *17.88 | 15.75 | 13.30 | 24.60 | 29.79 | 29.54 |
| **Ours** (unconditional) | *12.77 | 12.31 | 16.89 | 15.30 | 22.43 | 22.76 |
| **Ours** (conditional) | ***8.76** | **9.97** | 11.15 | **14.34** | **21.87** | **15.70** |

Table 2: Classification accuracy (%) on StandfordCars with semi-supervised learning evaluation protocols in the unified benchmark (Jiang et al., 2022).

| Label Set/Method | ERM | Pseudo Label | UDA | Fixmatch | Self Tuning | Flexmatch | Avg |
|---|---|---|---|---|---|---|---|
| 4 labels per category | 37.1 | 40.4 | 42.7 | 44.8 | **59.8** | 55.3 | 46.68 |
| + augmentation (ours) | **40.3** | **43.0** | **43.9** | **46.5** | 59.6 | **56.1** | **48.23** |

## 4.1 Evaluation Protocols and Implementation Details

To demonstrate the effectiveness of proposed tuning approach in natural image generation, we compare our approach with three GAN-based baselines: TransferGAN (Wang et al., 2018), Scratch-GAN, (Zhao et al., 2020), and two DPMs-based baselines: ScratchDiff, Tuning-All. TransferGAN initializes with the pretrained model (Mescheder et al., 2018) and fine-tunes the whole parameters. ScratchGAN trains GP-GAN (Mescheder et al., 2018) from scratch. (Zhao et al., 2020) investigates the transferable parts of pretrained GANs, and fine-tunes them with a further proposed adaptive filter modulation. As for the DPM-based methods, the insight of ScratchDiff and Tuning-All are similar to ScratchGAN and TransferGAN, respectively. For implementation, we adopt the latent diffusion model (LDM) (Rombach et al., 2022) pretrained on ImageNet with the resolution of 256×256. The hyper-parameters in LDM are retained. To obtain the semantic embeddings of inputs, we use the pretrained CLIP model (Radford et al., 2021) with ViT-B/32 backbone (Dosovitskiy et al., 2020). The base learning rate is set to 2.0e-6 and will be scaled up by the batch size and the number of GPUs. For sampling, we use the DDIM sampler (Song et al., 2020a) with 200 sampling timesteps. We use 4×RTX 3090 GPUs with a batch size of 8. Due to the fewer training parameters, our method can also be implemented on a single RTX 2080 Ti GPU with 11GB memory.

To validate the potential applications in data-scarce scenarios, we take the semi-supervised learning (SSL) as an example, which aims at learning a great recognition model with annotating few training samples. We randomly sample four labels per category, and then generate additional 10 samples for each labeled sample. The generated images will be involved into the labeled sample set. We evaluate it with the empirical risk minimization (ERM) baseline and five state-of-the-art SSL methods: Pseudo Label (Lee et al., 2013), UDA (Xie et al., 2020), FixMatch (Sohn et al., 2020), Self Tuning (Wang et al., 2021) and Flexmatch (Zhang et al., 2021).

## 4.2 Quantitative Results

**Quality of Generated Images.** We compare the quality of generated images by different approaches quantitatively, as measured by the widely used FID score (Heusel et al., 2017). For training with the full dataset, the FID score is computed w.r.t. the training set, as is standard practice (Ho et al., 2020). For training with a subset of the dataset, we compute the FID score on the full dataset. From Table 1, we have the following findings: (1) For the limited training data scenarios, fine-tuning the pretrained GP-GAN leads to mode collapse, while training the LDM from scratch fails to generate meaningful images with limited training steps. (2) Tuning the whole pretrained LDM is able to transfer the learned knowledge to a new task, and is comparable to the previous state-of-the-art (Zhao et al., 2020) on the limited data scenarios (i.e., CelebA-1k, Flowers-1k and StandfordCars-1k). (3) Our proposed tuning approach achieves the best FID scores on most cases, significantly outperforms other DPM-based methods with much less training parameters.

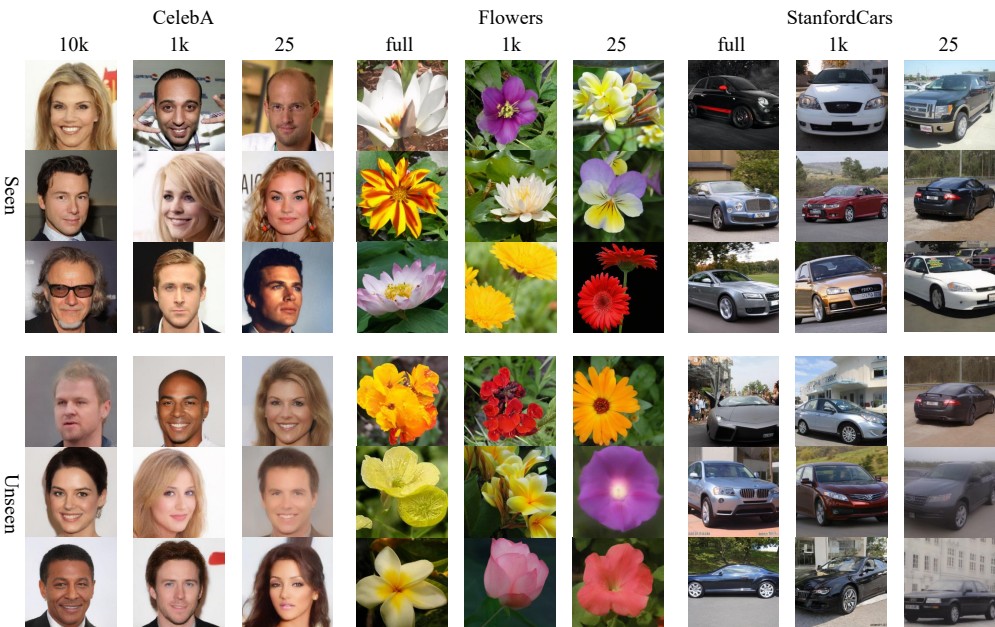

Figure 3: Images generated by our proposed tuning approach. We illustrate the results on three datasets: CelebA, Flowers and StandfordCars with different training sample sizes: full, 10k, 1k and 25. In the top three rows, the condition images come from the training set (i.e., is seen for the model). In the next three rows, the condition images are unseen for the model. Note that for Flowers-full and StandfordCars-full, all images are generated by the seen conditions.

**Ablation Study.** Our conditional model leverages the CLIP embeddings to achieve fast task transfer. To verify the effectiveness of this technique, we show the performance of unconditional model in Table 1. We can observe that the unconditional model performs slightly worse than the conditional ones since it lacks the semantic guidance. However, the unconditional model shows superiority on flexibility since it has no need of condition input during sampling. This ablation study reveals that the pre-trained encoder (e.g., CLIP) is not necessary for transferring pre-trained DPMs, but it can provide a noticeable boost. It is worth noticing that both unconditional and conditional models perform better than the baseline Tuning-All, demonstrating the efficacy of our proposed tuning approach.

**Tuning DPMs for Data Augmentation.** In this part, we empirically demonstrate the potential of our tuning approach in data augmentation. From Table 2, the augmented samples generated by our tuning approach improves the classification accuracy on five SSL methods. It is worth noticing that the augmented samples brings 3.2% and 2.6% improvement on the ERM and Pseudo Label, respectively. Compared to other SSL methods, the learning strategies of these two methods are simple, implying the effectiveness of the high-quality generated samples in data augmentation.

## 4.3 QUALITATIVE RESULTS

**Image Synthesis with Different Sample Sizes.** In Fig. 3, we show the generated images by our tuning method with different training sample sizes. We can observe that our method effectively transfers the pretrained DPM to generate with limited data. For CelebA-10k, Flowers-full and StandfordCars-full, the adapted model is able to generate high-quality images for both seen and unseen conditions. For small training size (e.g., CelebA-1k), the quality and reality of generated images are still competitive. As for learning with extremely limited data (e.g., 25) and sampling with unseen conditions, the generated images are slightly blurred, and the model conditioned with different images tends to generate similar images. This is because that the tuned model inevitably suffers from over-fitting when training with extremely limited data. However, when generating with seen conditions, our tuning approach is still effective with extremely limited training data. Another interesting finding is that the images generated with unseen conditions on CelebA usually focus on the faces and have blank backgrounds. More generated images are provided in Appendix C.

**Image Synthesis with Few Tuning Steps.** To validate the efficiency of our proposed tuning approach, we show the generated images with different tuning steps from 500 to 4000 with the interval of 500. From Fig. 1, we can see the coarse-to-fine adaptation process during tuning the pretrained LDM. In the beginning, our method learns the general outlines and shapes, and then supplements the details continually. From Fig. 4, we show the comparison between Tuning-All and our approach. We can find with less trainable parameters, our approach shows faster and better adaptation. It is worth noting that our tuning approach shows impressive generation quality with very limited iterations, which provides a convincing path for transferring pretrained DPMs to a new dataset with limited data and training resources.

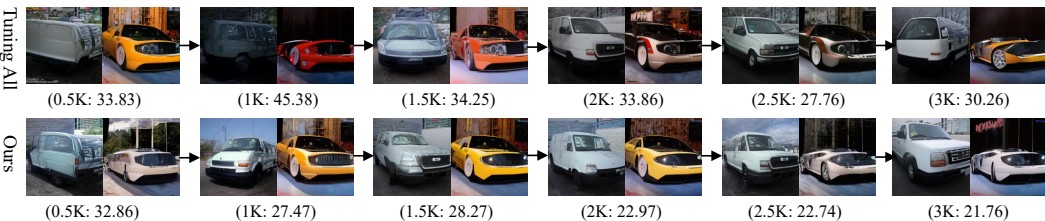

Figure 4: Generated images by Tuning-All and our approach with few tuning steps. To better validate our approach, the Tuning-All also utilizes the CLIP embeddings here. The first number under the image is the number of iterations, and the one after it is the FID score.

**Simple Text-to-Image Synthesis.** Recalling that we leverage the pretrained CLIP model to obtain the semantic embeddings of condition images, our method can achieve simple text-to-image synthesis since the CLIP embedding space is share by both vision and language modalities. We illustrate some language-prompted examples on the StandfordCars dataset in Fig. 5. We can observe that our method is able to generate suitable images w.r.t the given prompt texts like "a red car", "A black car driving on the road." and "There is a white car in front of the house". However, when given the prompt texts like "a truck", "a green car" or "A car driving in the rain.", the model fails to generate the corresponding images.

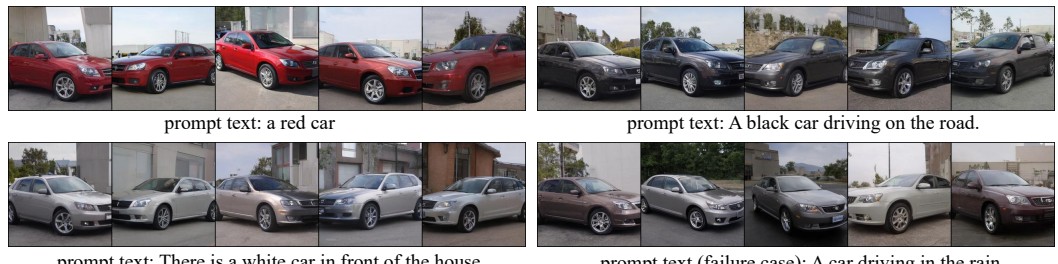

Figure 5: Illustration of simple text-to-image synthesis on the StandfordCars dataset.

**Masked Sampling.** We set the mask rate $\delta = 0.5$ (Sec. 3.3) and show the sampling results in Fig. 6. We can find that masked sampling promote the diversity of generated images when given the same condition image. Note that the mask rate is a trade-off between quality and diversity. When increasing the value of mask rate $\delta$, the diversity of generated images will be improved while the quality will be impaired. We find $\delta = 0.5$ is a great choice for balancing the diversity and quality. We provide detailed quantitative results in Appendix A.

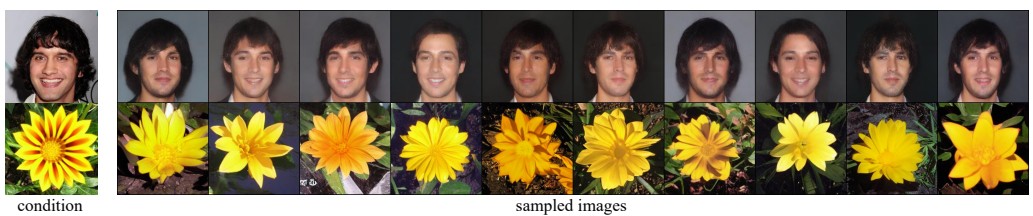

Figure 6: The results of diverse sampling.

## 5 RELATED WORKS

**Transfer Learning and Few-Shot Image Generation.** Transfer learning focuses on reusing learned knowledges from previous tasks for the learning of new tasks (Pan & Yang, 2009). For GAN-based image generation, TransferGAN (Wang et al., 2018) propose to fine-tune the whole model pretrained on source domain with limited target data. Zhao et al. (2020) follow the insights of transfer learning in discriminative tasks, and proposed to freeze the generally-applicable low-level filters of convolutional neural networks.

Another similar topic is few-shot image generation (Hong et al., 2020; Noguchi & Harada, 2019; Ojha et al., 2021; Liu et al., 2020). Despite that a few works are proposed, most of them focus on GANs, and infeasible for DPMs since they share totally different architectures and learning objectives. For DPM-based few-shot image generation, Sinha et al. (2021) uses a learned diffusion-based prior over the latent representations to improve generation. (Sinha et al., 2021) mainly addresses minor manipulations like generating faces with the blond hair or adding the beard for a male face. Recently, a concurrent work (Giannone et al., 2022) propose few-shot diffusion models by introducing a trainable vision transformer to condition diffusion models. It is worth noticing that (Giannone et al., 2022) trains an additional vision transformer and the whole diffusion models, while our method propose to take full advantages of pretrained models and only tunes the ANL modules.

**Diffusion Probabilistic Models.** Diffusion probabilistic models (DPMs) have been demonstrated the effectiveness in high-quality image synthesis (Ho et al., 2020). The DPM is first proposed by (Sohl-Dickstein et al., 2015) to generate by reversing a diffusion process using a Markov chain with discrete timesteps. DDPM (Ho et al., 2020) leverages noise prediction networks to parameterize the mean of the Markov chain, in which different timesteps share the same parameters. To improve DDPM, various approaches are proposed from different aspects, such as learning on latent space (Pandey et al., 2022; Rombach et al., 2022), faster sampling (Song et al., 2020a; Liu et al., 2022). Song et al. (2020b) propose that the DPM with infinitesimal timesteps can be representation by stochastic differential equations (SDEs). To improve the score-based diffusion models, Song et al. (2021) further propose likelihood-based training scheme, and lots of efforts are paid for faster sampling like designing faster SDEs solvers (Jolicoeur-Martineau et al., 2021; Popov et al., 2021) or analytically estimate the optimal reverse variance (Bao et al., 2022a;b).

## 6 LIMITATIONS AND SOCIETAL IMPACT

**Limitations.** While our proposed tuning approach significantly reduces optimization steps and the demand for training data, it fails to generate high-quality images with unseen conditions when extremely limited data (e.g., 25) are provided. Another limitation is that although our approach reduces the training parameters from 400M (conditional LDM pretrained on ImageNet with the resolution of 256×256) to 169M, 169M trainable parameters are still a little bit large. The parameters mainly come from the non-linear mapping module which consists two fully-connected layers and an activation layer. Further reducing the parameters that need tuning is a direction for future works.

**Societal Impact.** Our paper presents a general transfer learning approach for DPMs, so it may help expand any impact that generative models had on the broader world. For natural image generation, generative models may encounter biased, deliberate synthesis and copyright infringement issues. Our method enables transferring a pretrained DPM to a specific task using much less data and computing resources. We hope that our approach can be used to achieve various creative and positive generations, such as synthesizing early prototypes for artists, or augmenting and extending datasets for the data-scarce scenarios (Antoniou et al., 2017).

## 7 DISCUSSIONS

We introduce a new tuning approach for leveraging pretrained DPMs to generate high-quality natural images with limited training data and optimization steps. The advances in DPMs such as faster sampling algorithms can be used to further improve our tuned model since our method are orthogonal to them. To the best of our knowledge, we are at the very early attempt to investigate transfer learning in diffusion probabilistic models. We hope our work paves the way for small laboratories and companies as well as individual researchers to train task-specific generative models.

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

## A    RESULTS OF DIFFERENT MASK RATE

In Sec. 3.3, we propose a simple masked sampling approach for conditional diverse sampling. Intuitively speaking, the masked sampling will increase the diversity of generated samples, but hurt the quality. To verify it, we use the FID score to evaluate the quality, and the Inception Score (IS) to evaluate the diversity, respectively. The results are shown in Table 3. Since the StandfordCars is a single-class dataset, the inception score will always keep a small value. From Table 3, we can find the FID and IS scores increase by enlarging the mask rate, revealing that there exists a trade-off between the quality and diversity of generated images.

Table 3: The FID and IS scores of the generated images with different mask rate. The evaluated model is trained on StandfordCars dataset with 4k tuning iterations.

| mask rate | 0% | 10% | 20% | 30% | 40% | 50% | 60% | 70% | 80% | 90% |
|---|---|---|---|---|---|---|---|---|---|---|
| IS $\uparrow$ | 3.18 | 3.14 | 3.10 | 3.18 | 3.25 | 3.46 | 3.56 | 3.87 | 4.12 | 4.21 |
| FID $\downarrow$ | 22.74 | 23.23 | 22.96 | 24.08 | 25.08 | 27.28 | 35.08 | 44.20 | 55.86 | 68.38 |

## B    GENERALIZE TO OTHER DPMS

In this paper, we focus on natural image generation with latent diffusion models (LDM). Here, we provide intuitions for other DPMs like DDPM. Unlike the pre-trained LDM, the pre-trained DDPM does not contain the ANL module, thus may introduce more optimization steps since the ANL module will be randomly initialized. However, since LDM inserts the ANL module into each residual convolutional block of U-Net, it is not convenient to further reduce the number of layers that needed to be tuned. For DDPM, we suggest to discover a more efficient insert pattern to further reduce the tuning parameters. Another interesting direction is to extend our method to other generative tasks like text-to-image generation, text-to-speech synthesis and protein structure generation. For example, stable diffusion model (Rombach et al., 2022) has achieved impressive performance on text-to-image generation, and our method can be used to transfer it to a specific domain.

## C    ADDITIONAL SAMPLES

In Fig. 3, we have shown some images sampled by different models. Here, we provide more generated images with different conditions in Fig. 7, 8, 9, 10, 11, 12, 13, 14, 15 and 16. To avoid the pdf file being too large, we compressed these images to 96 dpi.

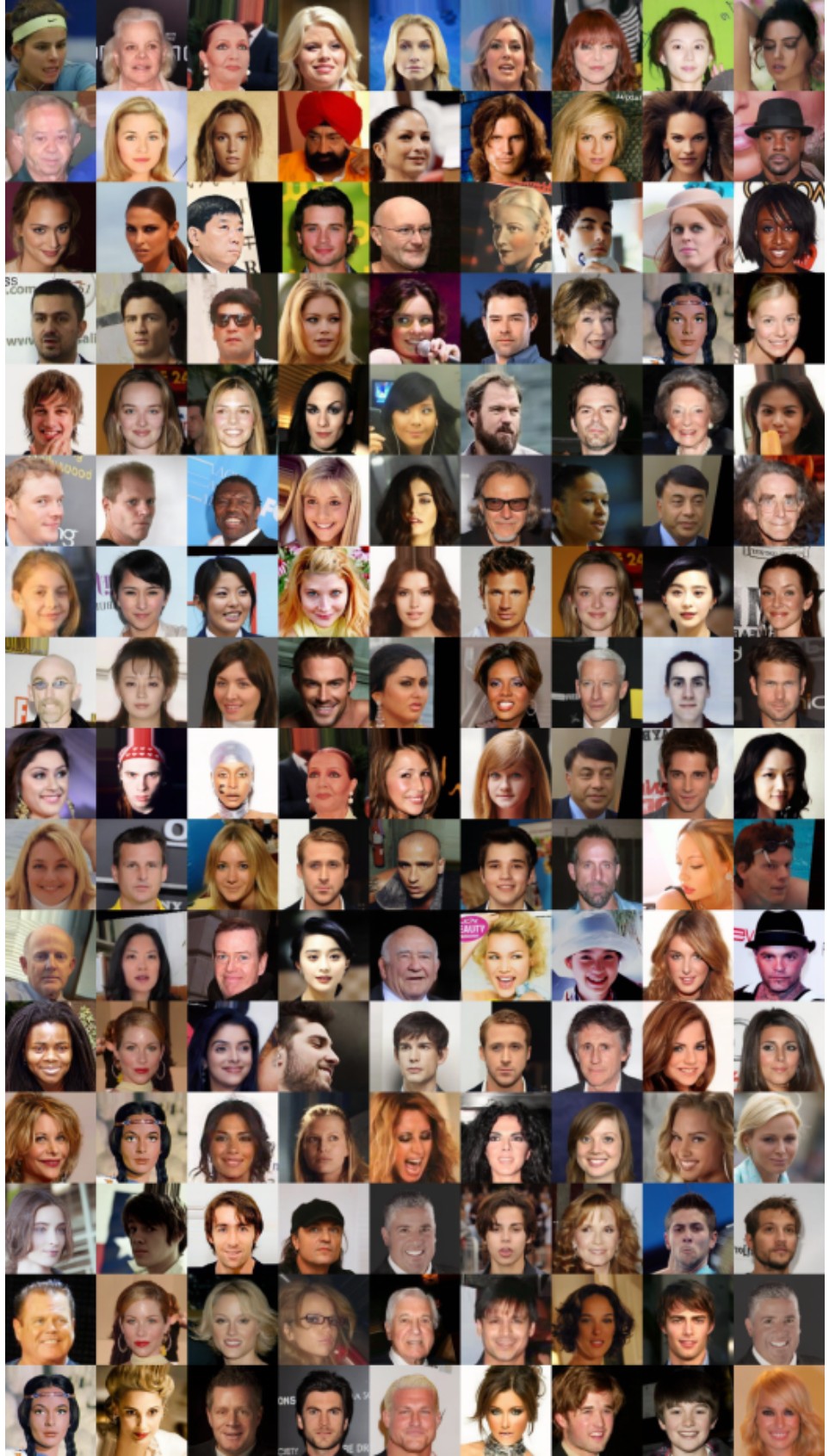

Figure 7: The 256×256 sampled images on CelebA-10k with seen condition images.

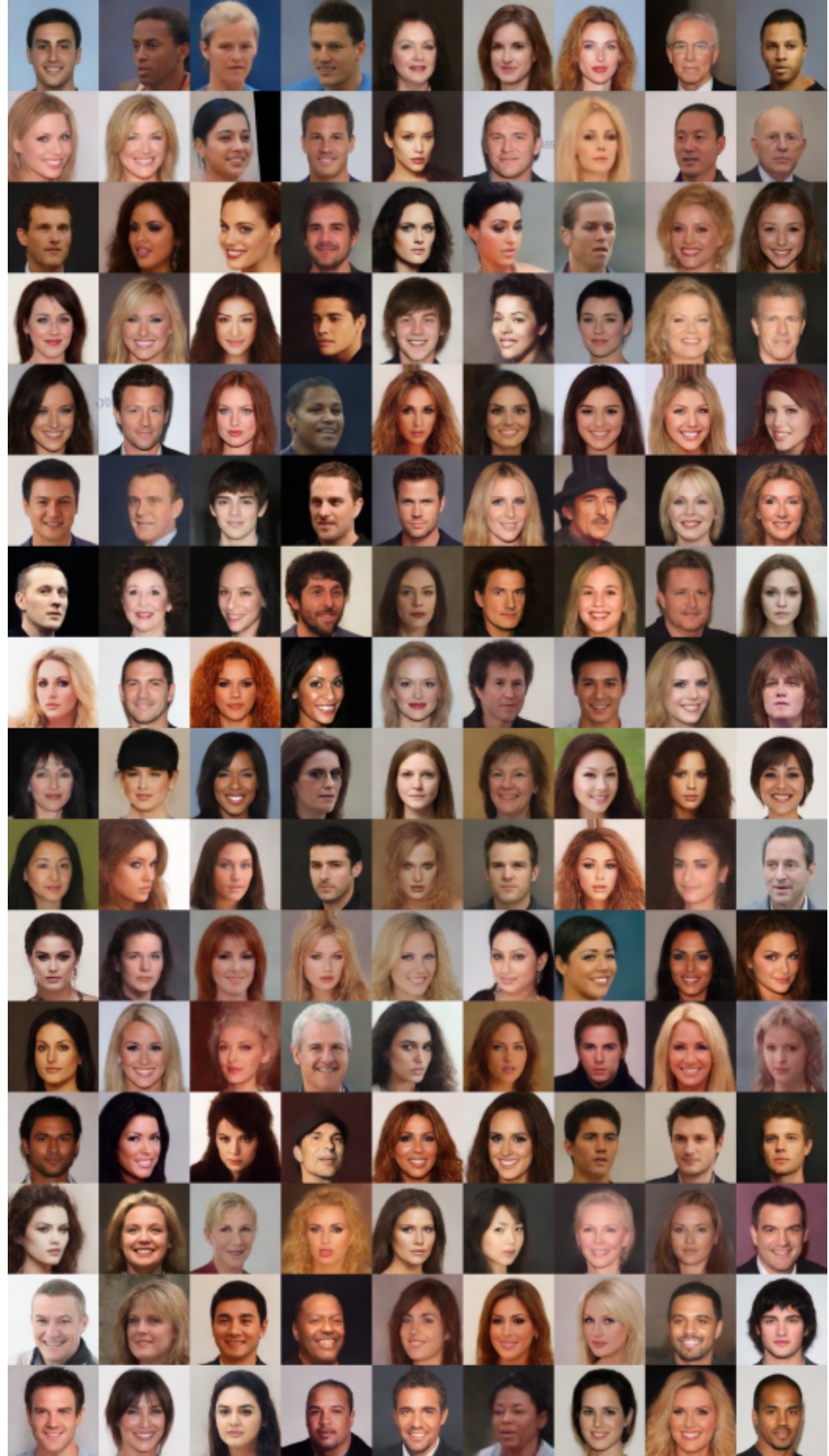

Figure 8: The 256×256 sampled images on CelebA-10k with unseen condition images.

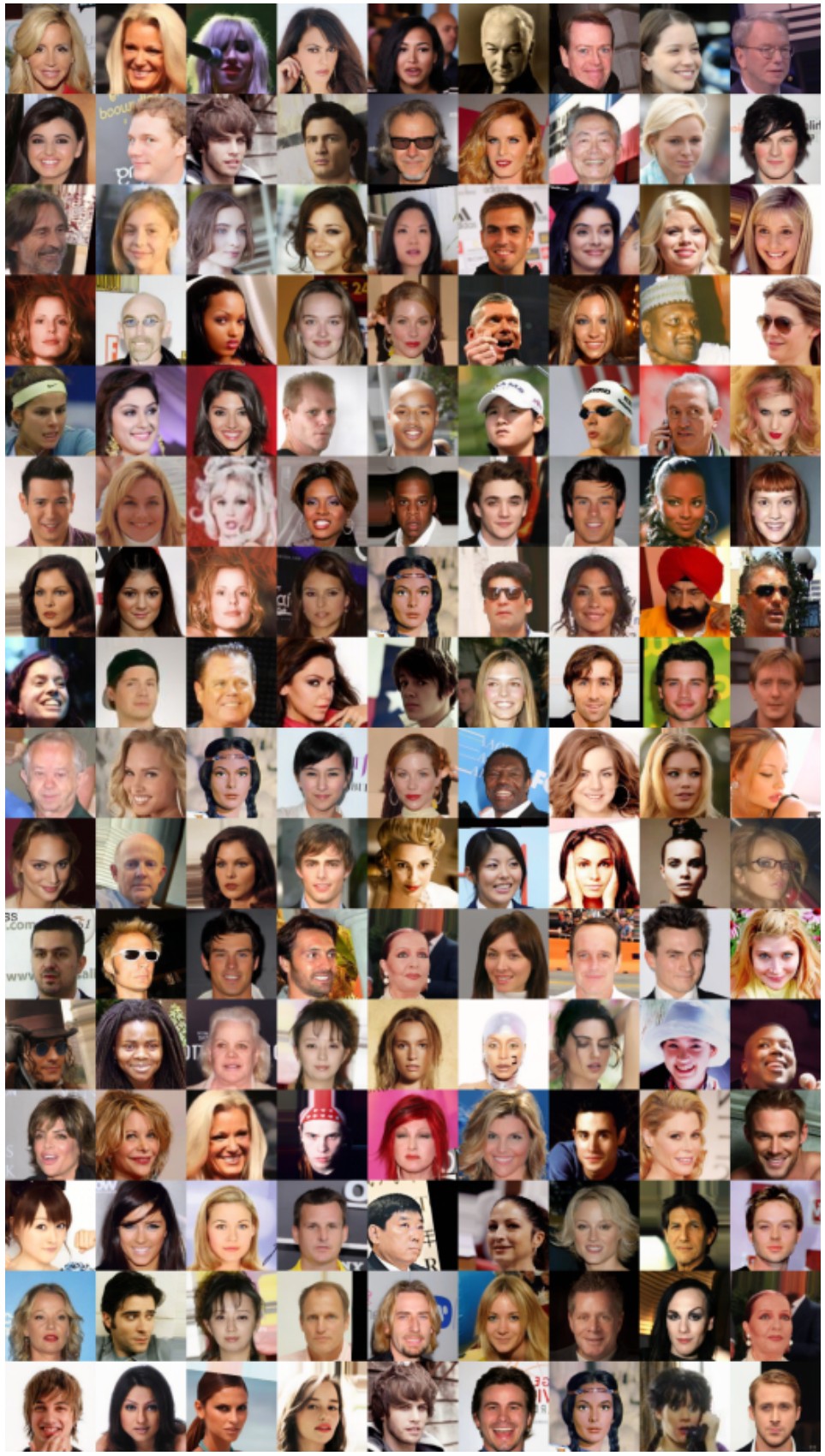

Figure 9: The 256×256 sampled images on CelebA-1k with seen condition images.

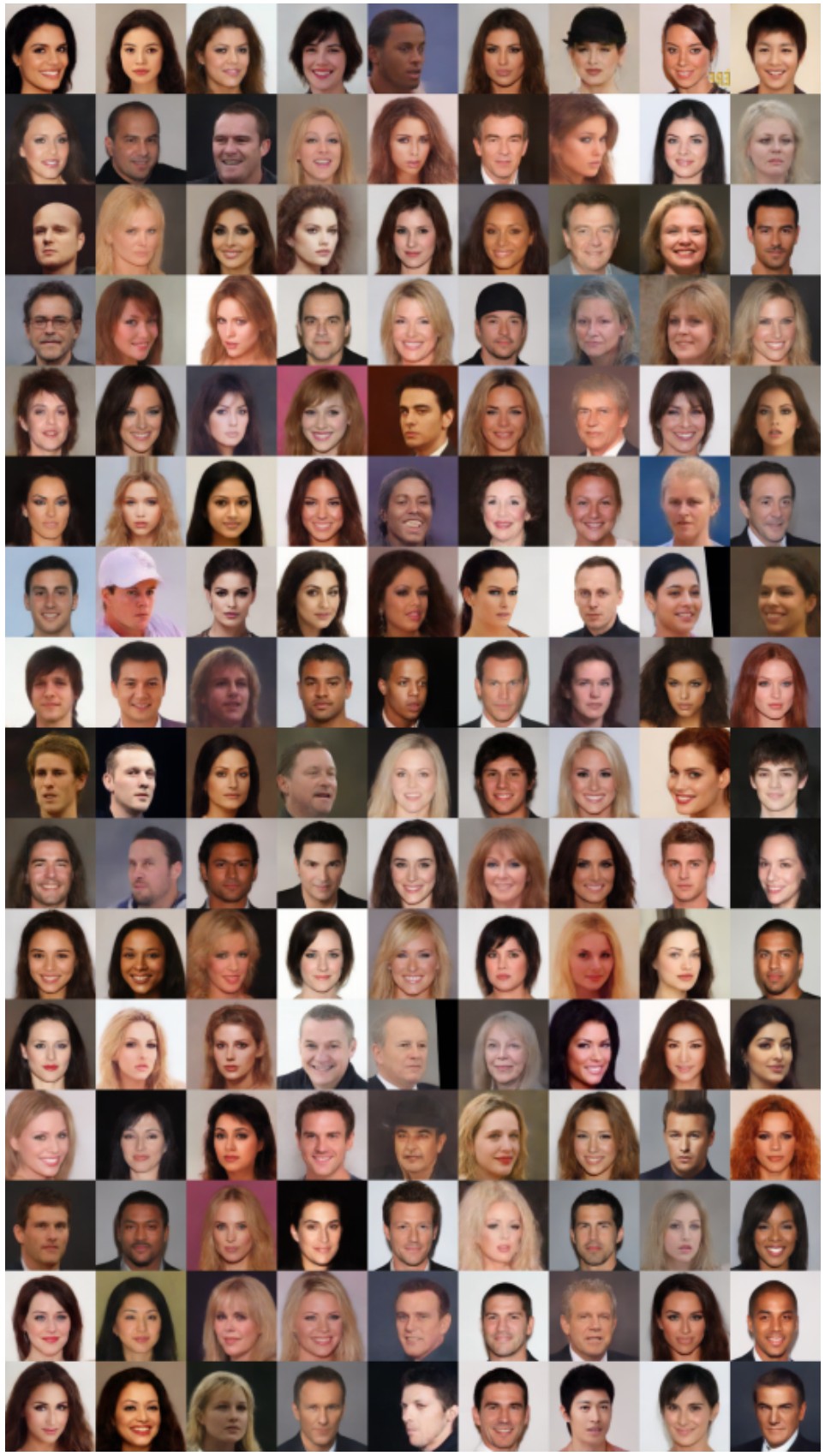

Figure 10: The 256×256 sampled images on CelebA-1k with unseen condition images.

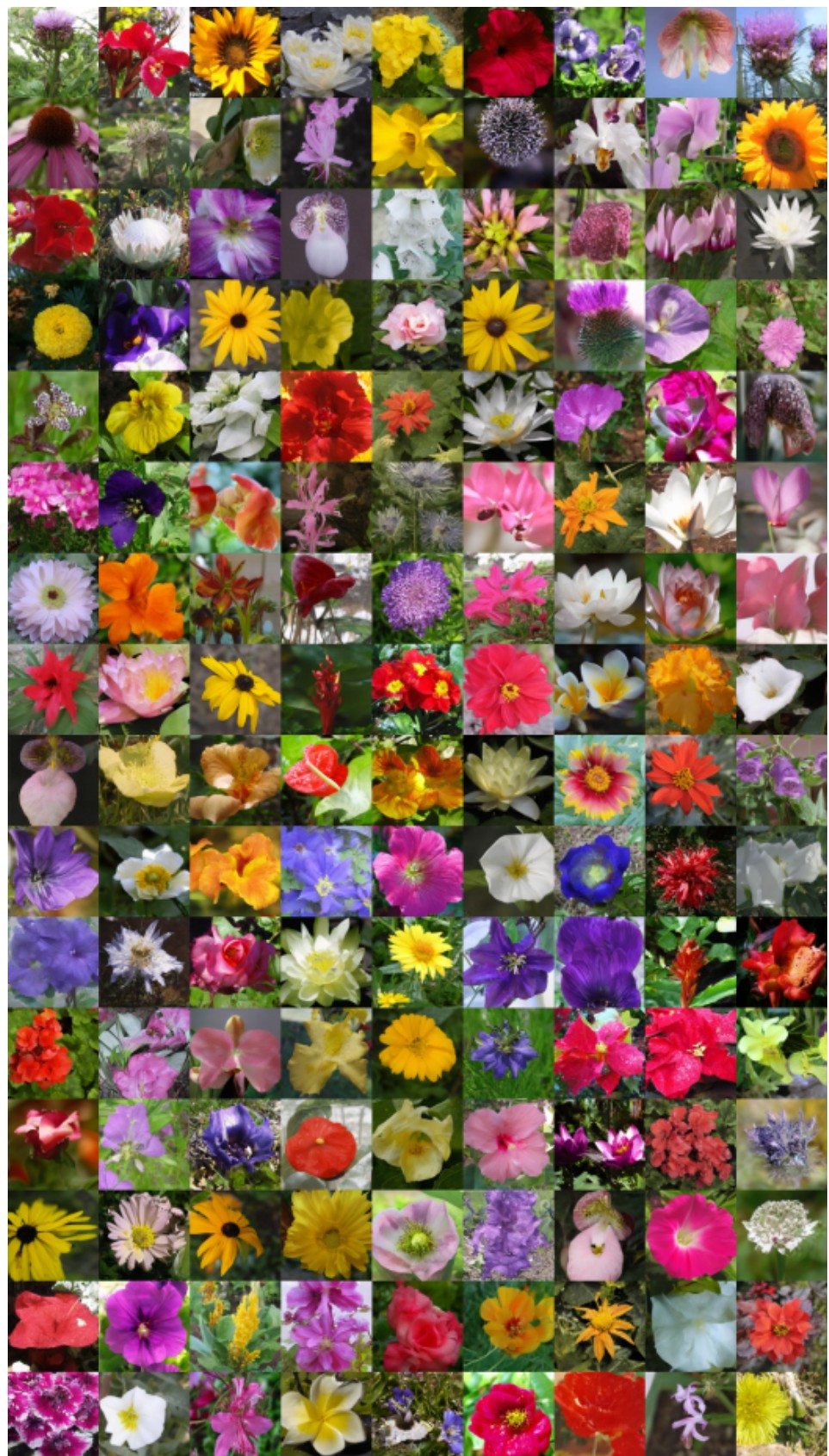

Figure 11: The 256×256 sampled images on Flowers-full.

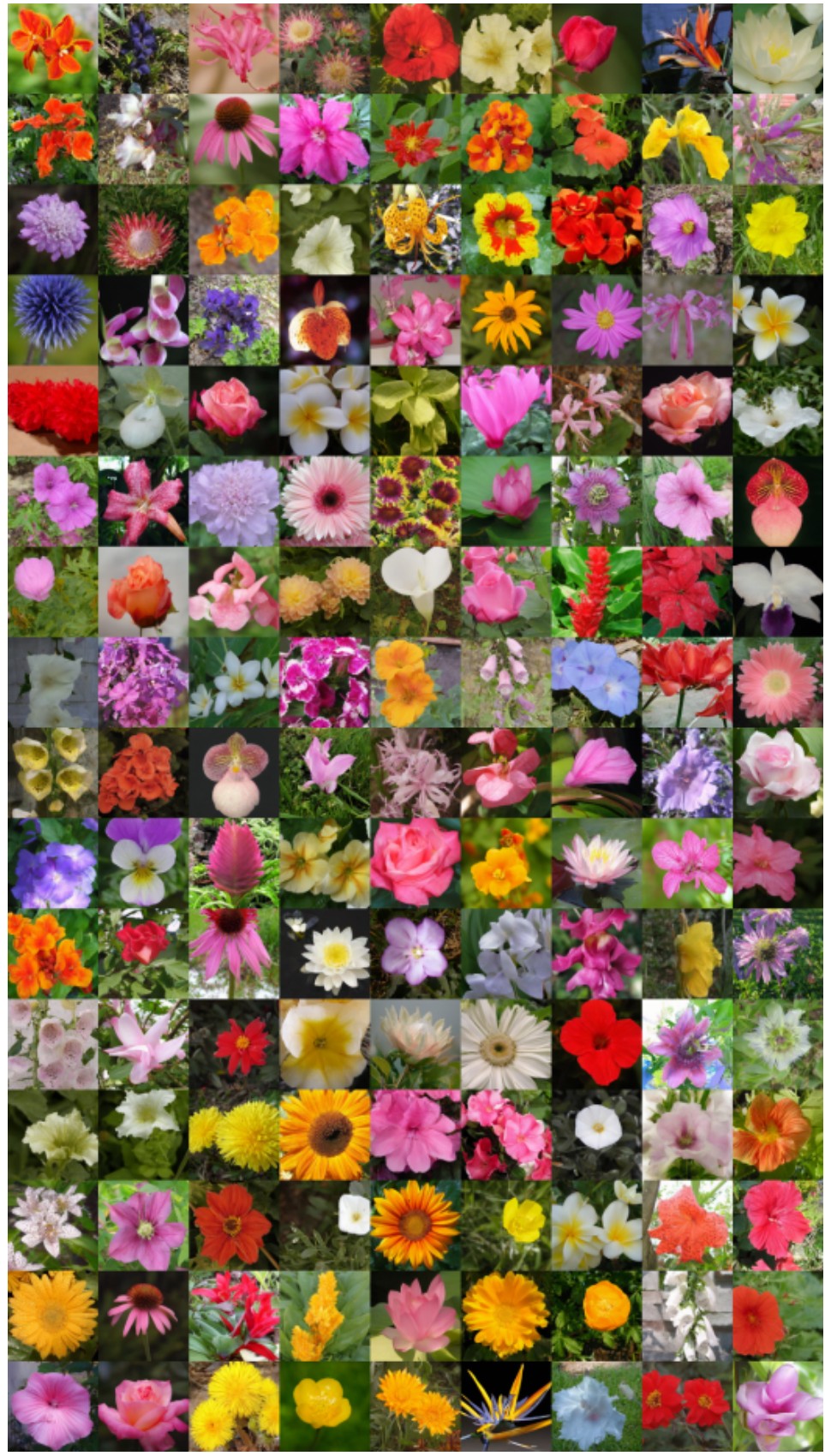

Figure 12: The 256×256 sampled images on Flowers-1k with seen condition images.

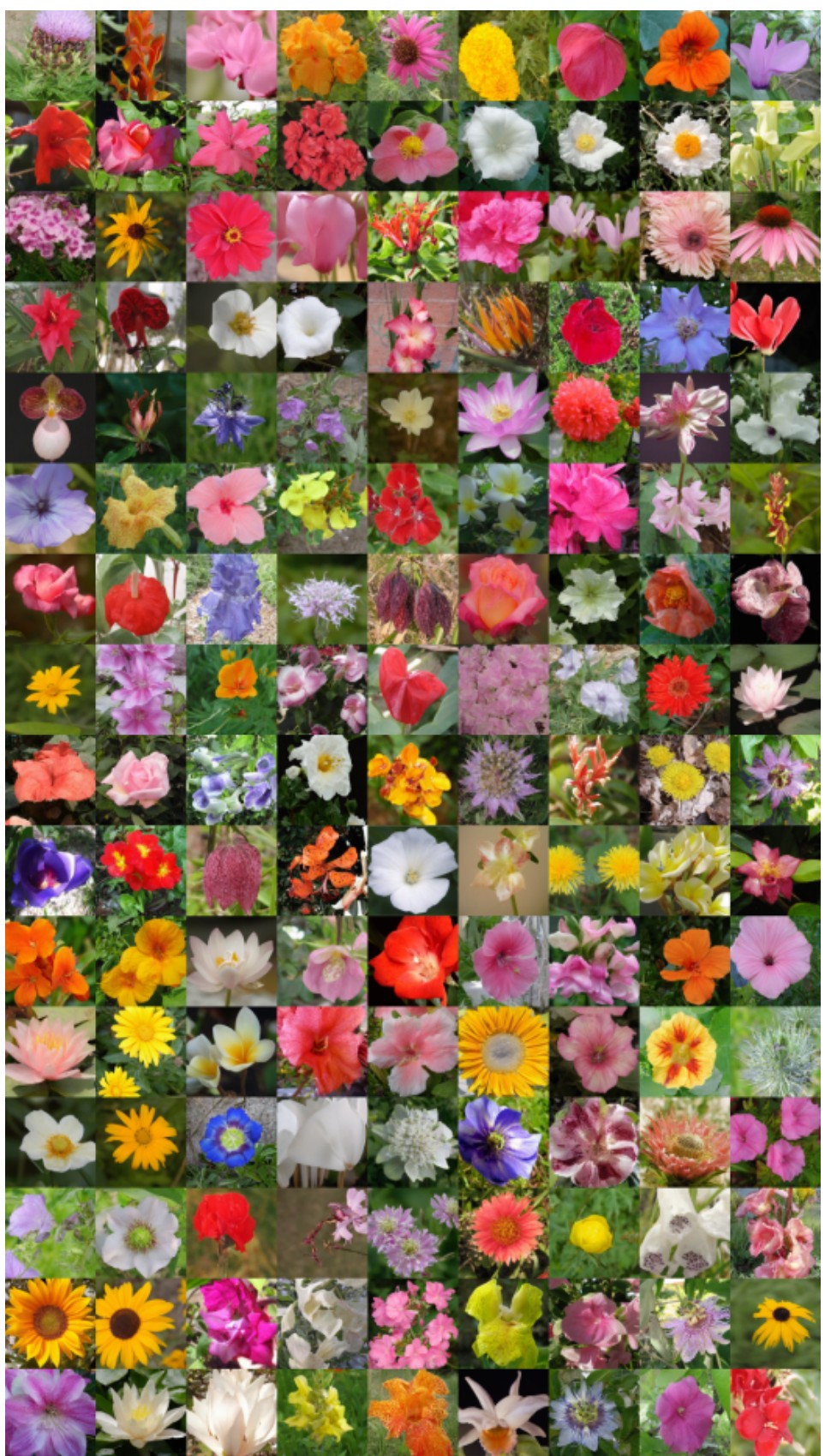

Figure 13: The 256×256 sampled images on Flowers-1k with unseen condition images.

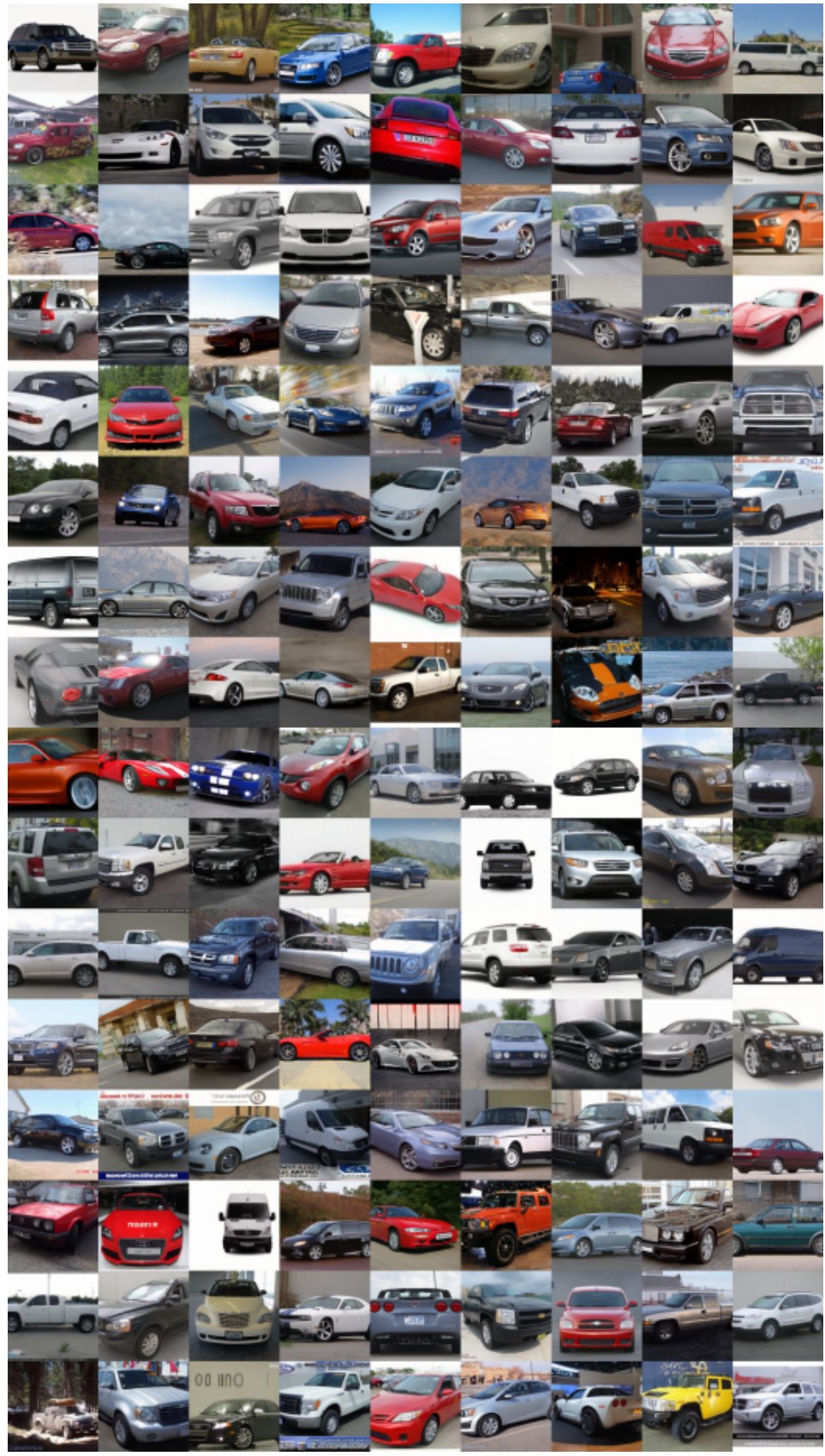

Figure 14: The 256×256 sampled images on Cars-full.

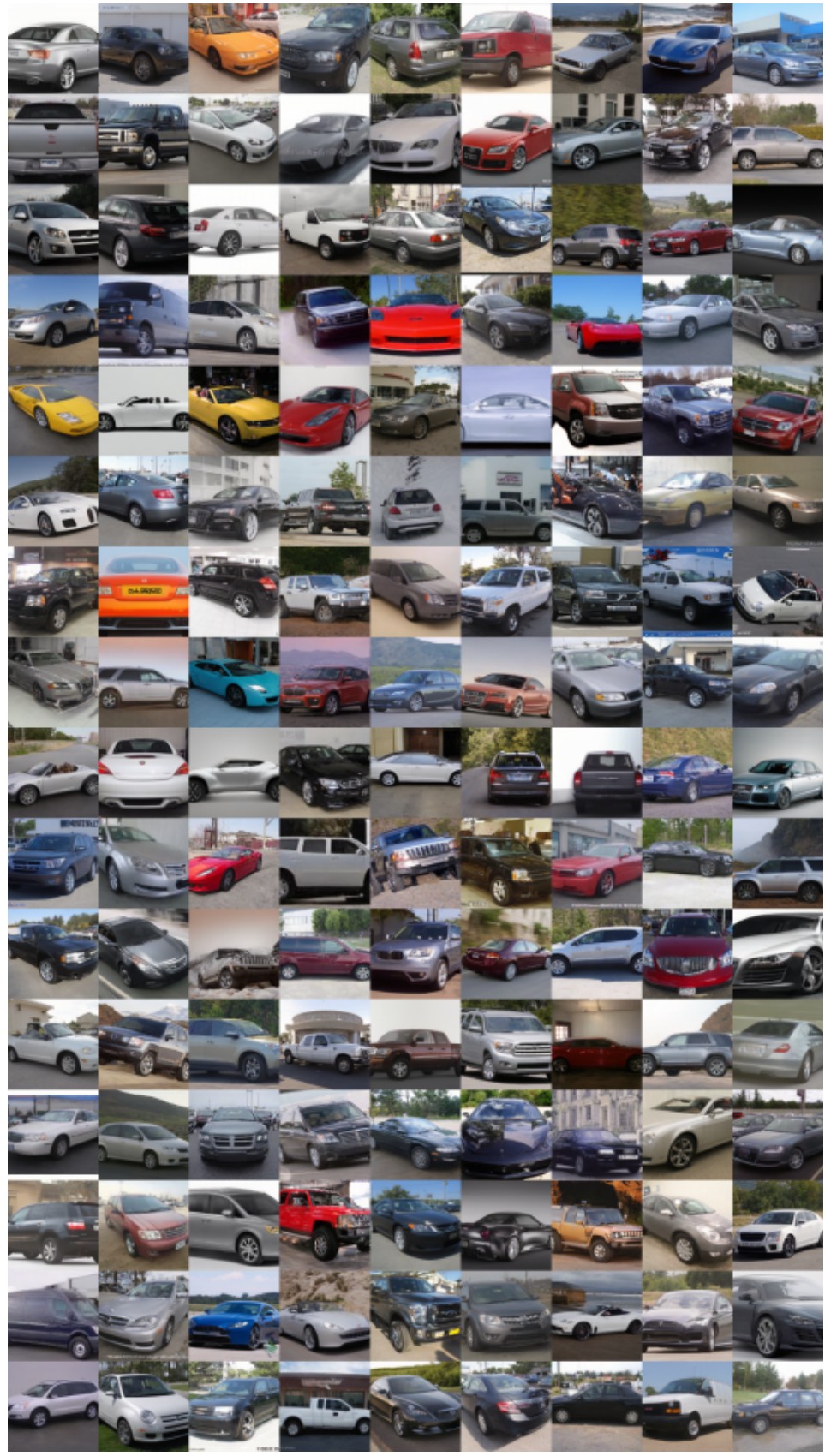

Figure 15: The 256×256 sampled images on Cars-1k with seen condition images.

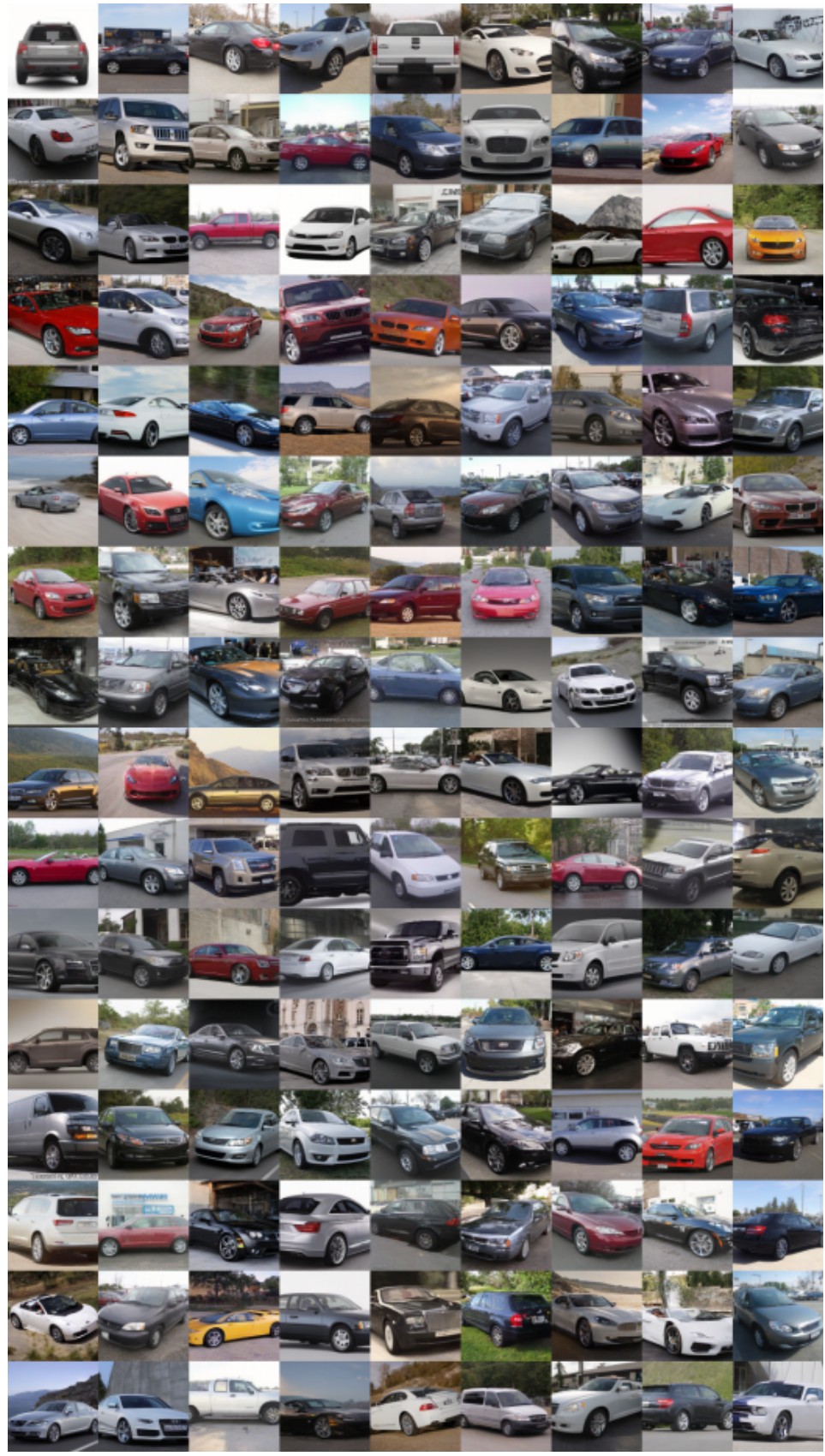

Figure 16: The 256×256 sampled images on Cars-1k with unseen condition images.

