# OpenReview forum: "Transferring Pretrained Diffusion Probabilistic Models"
_ICLR.cc/2023/Conference — Submitted to ICLR 2023_

### Official Review · Reviewer_wtTC · 2022-10-22

**Confidence:** 4
**Correctness:** 3
**Technical Novelty And Significance:** 3
**Empirical Novelty And Significance:** 2
**Recommendation:** 5

**Clarity, Quality, Novelty And Reproducibility:**


Clarity: 9/10

Quality: 7/10

Novelty: 6/10

Reproducibility: 5/10


**Strength And Weaknesses:**

Pros:

- The paper is well written and easy to follow.

- The proposed method is well motivated and convincing in real-life scenarios, as the training of diffusion models is really time-consuming.

- The experimental result is solid compared with related baselines.


Cons:

-"We investigate transfer learning in recent DPMs, and uncover that previous methods like training from scratch or determining the transferable parts are not effective.", I am doubtful about this statement in contribution and the failed result in Table 1, maybe running DDPM instead of LDM cannot fail? Failure on LDM is expected, as you cannot update the Encoder-Decoder in LDM in your fine-tuning, any sensitive parameter updating inside will greatly affect the sampling quality of LDM.

- ANL module is not novel, already been proposed in latent diffusion models, etc.

- Contribution 2 is not clear, efficient on what? speed? parameter number? convergence?

- Lacking in-depth analysis of ANL modules, which layer should we insert the ANL module? Should we insert it into all layers by default? Maybe we can find a more efficient inserting pattern?

- Lacking in analysis of overfitting problem, the models will meet the problem of catastrophic forgetting so that it fails to generate samples from old distribution.

- In Figure 2, your model's trainable parameter number is 169M for few images(1k)? This should inevitably lead to an overfitting problem, as mentioned above.

- In sampling, you still need the CLIP embedding, where is the clip-embedding comes from? How masked sampling performs to increase the diversity of the generated samples (the effect of Fig6 can be quantified by Inception Score as function of mask rate.)? As CLIP embedding is a too-fine-grained condition so that the sampling might only mesmerize the training data.

- The exposition of masked sampling is too abrupt, the motivation is missing there.

- FID/IS of generated samples should be compared before and after deploying the masking, I think there will be trade-off issue between FID and IS.

- Details about the "tuning DPMs for data augmentation" is missing, how to obtain the condition of CLIP embedding during this process of sampling? How many samples are used in this experiment?






**Summary Of The Paper:**

This paper focuses on exploration of transfer learning in diffusion models, technically, they propose an Attention-NonLinear (ANL) module to facilitate the conditioning process of CLIP embedding. To avoid the diffusion model overfits into the clip embedding of the images, they also generated an approximate clip embedding from masked images.

The experimental results show that the adaptation can be achieved by few-training data and steps, and outperforms related baselines. Also, the generated samples can further boost the performance of semi-supervised learning.

**Summary Of The Review:**

This paper is well motivated, while lacking in some in-depth analysis around the proposed contributions. Some details are missing to hinder the reproducibility of the paper.

---

> ### Author Response · Authors · 2022-11-11
> **Response to Reviewer #wtTC (Part 1/2)**
>
> We appreciate for acknowledging the motivation and convincing applications of our work. Here, we address your concerns as follows:
>
> **Q1**: Ambiguous statement in Contribution 1.
>
> **A1:** Thanks for your suggestions. We agree with you that training DDPM from scratch with lots of iterations (e.g., 60k in Table 1) can generate meaningful images. However, it is hard to train DDPM from scratch with limited iterations. For instance, we train the DDPM on StandfordCars with default configurations (https://github.com/lucidrains/denoising-diffusion-pytorch) except for the image size (256) and DDIM steps (200) for fair comparison, and find that training DDPM from scratch with limited iterations (e.g., 3k) fails to generate meaningful images, while our tuning method can already generate high-quality images (see Figure 1 and 4). We would like to include the full results of DDPM in our revised version, but it needs time for implementation.
>
> **Q2:** ANL module is not novel.
>
> **A2:** Our contribution is not to propose this module, but to propose an effective tuning method.
>
> **Q3:** Contribution 2 is not clear.
>
> **A3:** Thanks for your suggestions. Our tuning approach has less parameters needed to be tuned and converges faster than the baseline “Tuning All”. We will revise the unclear statement.
>
> **Q4:** Can we find a more efficient inserting pattern?
>
> **A4:** The reviewer raises an interesting point. Firstly, we want to clarify that the ANL module is not inserted into all layers, but is inserted into each residual convolutional block of U-Net. Since the pre-trained model comes from latent diffusion models, whose attention modules are organized as this pattern, we thus preserve this structure. We believe that exploring a more efficient inserting pattern, or more generally, further reducing the amount of parameters is an import direction for future work.
>
> **Q5:** About the overfitting and catastrophic forgetting problem.
>
> **A5:** Firstly, learning with limited samples will inevitably lead to an overfitting problem. Since we use the LDM pre-trained on ImageNet with 256 $\times$ 256 resolution, the number of parameters is larger than the models trained on the datasets like CIFAR (32 $\times$ 32) or CelebA (64 $\times$ 64). However, with enough iterations, our model performs better than the baseline that tuning the whole parameters (see Table 1), revealing that our method alleviates the overfitting problem when compared to the baseline. Note that the key contribution of our paper is a new tuning approach, which also can be applied to other lightweight pretrained diffusion models to reduce the parameters. The catastrophic forgetting problem is often mentioned in the incremental/continual learning community, in which the new distribution is usually constrained (e.g., learn a new category based on the source model). However, in this paper, we consider the transfer learning problem, in which the gap between old and new distribution is really large. For instance, the marginal ($p(x)$) and conditional ($p(y|x)$) distribution between ImageNet and CelebA are totally different. Therefore, transfer learning focuses on the new (target) distribution, the performance on the pre-training dataset is not considered.
>
> **Q6:** Some concerns on sampling and the masked sampling section is abrupt.
>
> **A6:** In sampling, the CLIP embedding comes from the given images, which might affect the diversity of generated images as mentioned in your reviews. This issue motivates us to provide a simple masking strategy for sampling. We will revise the statement to improve readibility. It is worth noticing that our proposed tuning approach can be also applied to unconditional generation, in which there is no given prompt during sampling (see Appendix A in our original submission).

---

> > ### Author Response · Authors · 2022-11-11
> > **Response to Reviewer #wtTC (Part 2/2)**
> >
> > **Q7:** Comparison between the samples before and after deploying the masking.
> >
> > **A7:** Thanks for your suggestions. Here, we show the FID and IS scores of generated samples before and after deploying the mask on StandfordCars dataset. Since the rebuttal time is limited, we report the results after 4k tuning iterations, and generate 1k images for each variant.
> >
> > | mask rate | 0% | 10% | 20% | 30% | 40% | 50% | 60% | 70% | 80% | 90% |
> > | ---- | ---- | ---- | ---- | ---- | ---- | ---- | ---- | ---- | ---- | ---- |
> > | IS $\uparrow$ | 3.18$\pm$0.17 | 3.14$\pm$0.23 | 3.1$\pm$0.17 | 3.18$\pm$0.22 | 3.25$\pm$0.16 | 3.46$\pm$0.22 | 3.56$\pm$0.18 | 3.87$\pm$0.26 | 4.12$\pm$0.36 | 4.21$\pm$0.37 |
> > | FID $\downarrow$ | 22.74 | 23.23 | 22.96 | 24.08 | 25.08| 27.28 | 35.08 | 44.20 | 55.86 | 68.38 |
> >
> > As expected, there exists a trade-off between FID and IS. However, we have to emphasize that the Inception Score is not a good metric for our experiments since we conduct experiments on single-label dataset, which always gains a low Inception Score. The Inception Score is more suitable for the generative tasks on the datasets like ImageNet and MS-COCO.
> >
> > **Q8:** Details of the experiment on semi-supervised learning.
> >
> > **A8:** We generate the augmented samples with the condition of labeled data, which provides the CLIP embeddings. As for the portion of labeled samples, we follow the evaluation protocols in this benchmark [1] (mentioned in Table 2), randomly sample four labels per category.
> >
> > Thanks again for your valuable reviews. Please let us know if you have any further questions on our work.
> >
> > Reference:
> >
> > [1] Jiang et al, Transferability in deep learning: A survey, Arxiv’2022

---

> > ### Author Response · Authors · 2022-12-11
> > **A kind reminder**
> >
> > Dear Reviewer #wtTC,
> >
> > Thanks again for all the time and effort in reviewing our paper. As you might know that the reviewer final recommendation deadline is approaching. We appreciate your valuable comments and have made detailed responses to them. We have also incorporated all your valuable suggestions to our revised manuscript.
> >
> > During the discussion phase between reviewers and AC, we added two experiments to show that (1) Our tuning approach shows better and more stable performance than the baseline with different learning rates. (2) Our tuning approach enables effective transfer learning while freezing the encoder (i.e., only tuning the block9-16) of pre-trained U-Net, which provides a insight to further reducing the tuned parameters.
> >
> > If our response has addressed your concerns, we would highly appreciate it if you could re-evaluate our work and consider raising the score. Also, we are happy to engage in discussions if you have any further questions on our work.
> >
> > Best regards,
> >
> > Authors of Paper #2648

---

### Official Review · Reviewer_uRV2 · 2022-10-24

**Confidence:** 4
**Correctness:** 3
**Technical Novelty And Significance:** 3
**Empirical Novelty And Significance:** 3
**Recommendation:** 5

**Clarity, Quality, Novelty And Reproducibility:**

- The paper is clearly written and easy to understand. I believe the proposed approach is novel and technically sound. The details to reproduce the experiments in the submission are provided in the paper and all the datasets are publicly availalble.

**Strength And Weaknesses:**

Strengths:
- The problem of fine-tuning large pre-trained models on small datasets in an efficient way is interesting and deserves exposure in the community.
- The proposed approach is technically correct and the empirical results support the main claims in the paper.

Weaknesses:
- There’s one important baseline missing. That is figuring out how important the semantic latent c is for this approach to work. In other words, would this work with latents that do not come from big models like CLIP? I suggest the authors run the following experiments to better understand this phenomenon: - 1) Use a CLIP visual encoder with random weights and fine tune it on the small dataset. - 2) Let the latent c of each example be non-parametric (eg. a nn.Embedding in pytorch). In this way, one will let the fine-tuning approach to backprop all the way to these non-parametric latents.

**Summary Of The Paper:**

This submission presents a method for fine-tuning large generative models on small datasets efficiently. The proposed approach used CLIP to encode images from the small datasets into a semantic latents space that is then used to condition the frozen pre-trained model via a cross-attention module. The approach is evaluated on several standard benchmarks datasets outperforming the baselines where models are trained from scratch or where all the parameters of the pre-trained model are fine-tuned.

**Summary Of The Review:**

This paper proposes an approach for fine-tuning large pre-trained generative models on small datasets in an efficient way. I believe this problem is interesting for the community and deserves exposure. The proposed approach is technically correct and the empirical evaluation is sound. However, I believe there’s a couple of major baselines that need to be run in order to better understand the contributions (see weaknesses).

---

> ### Author Response · Authors · 2022-11-11
> **Response to Reviewer #uRV2**
>
> We appreciate for acknowledging the novelty and technical soundness of our work. The answer to the question that how important the semantic latent is for our approach is already provided in our original submission (see **Appendix A**). In Appendix A, we validate our tuning approach in unconditional generation, in which the semantic latent from CLIP is removed. In the low-data regime (e.g., 1k training samples), this variant performs slightly worse than the CLIP-conditioned ones. We want to emphasize that the tuning approach is the key technique for transferring pre-trained DPMs to data-scarce scenarios, the CLIP embedding is used to further improve it. Based on this, we discuss the mentioned two baselines: (A) random initialized CLIP embedder and (B) non-parametric embedder. Firstly, the motivation of this paper is to reduce the tuning parameters and optimization steps for small datasets, introducing additional parameters for just learning a semantic latent is unwise. In fact, in our early experiments, we have tried to learn the embedding by a ViT motivated by [1]. We find that this variant is unstable and performs badly for transferring pre-trained DPM to small datasets. In short, pre-trained CLIP $\approx$ pre-trained ViT > learnable random-initialized embedders. Secondly, the non-parametric embedder is similar to our experiment in Appendix A, which also removes the effect of semantic latent. This variant performs slightly worse than the conditional model, which benefits from the guidance of semantic latent. This ablation reveals that it is the proposed tuning approach rather than the CLIP embedding that is important for transferring pre-trained diffusion models.
>
> Thanks again for your valuable reviews. Please let us know if you have any further question on our work.
>
> Reference:
>
> [1] Giannone et al, Few-shot diffusion models. Arxiv’2022

---

> > ### Comment · Reviewer_uRV2 · 2022-11-15
> > **Continuing discussion of pre-trained embedder**
> >
> > Thanks for engaging in the discussion. As it currently stands I understand that for transferring pre-trained DPMs to small datasets one needs to have a pre-trained encoder (either CLIP or ViT) to get maximum performance. I think it would be good for authors to be upfront about the data required for training these models which they assume are given. This should be pointed out early on in the paper.
> >
> > The explanation of the ablation in Appendix A is not completely clear to the reviewer, it is hard to interpret what parts of the model are fine-tuned. Do you mean that the weights that are updated are only the ones in the self-attention layers of the particular DPM implementation? (in this case an LDM). There's no mention in the paper of a self-attention module (the first time it is mentioned it's on Appendix A), I think it would be good to clarify this and perhaps show these experiments early on in the paper.  If I'm interpreting the results on Appendix A correctly, the seem to point out that pre-trained encoders are not really required for the approach to work (but they certainly provide a noticeable boost), which I think should be highlighted early on in the paper. Additionally, I think readers would benefit from having a visual comparison of the results in Tab. 3. to strengthen the point that CLIP is not required.
> >
> > Finally, it seems that the proposed approach is specifically tailored for LDMs. I would suggest that authors include a section in the appendix providing intuitions or future directions to design approaches that are agnostic of the particular DPM architecture.

---

> > > ### Author Response · Authors · 2022-11-15
> > > **Thanks for your suggestions. & A clarification of Appendix A.**
> > >
> > > Thanks for your continued attention on our work. Here, we want to clarify a misunderstanding of Appendix A firstly. In Appendix A, we mention that “Our tuning approach can be also applied to the unconditional pretrained DPMs by replacing the cross-attention module with the self-attention module”. Note that the ANL module consists an attention layer and a non-linear mapping module. In other words, we tune the self-attention layers and non-linear mapping layers to achieve task transfer. The pre-trained encoders are not necessary, but they can provide a noticeable boost as you said. The key contribution of our paper is that, we present a new tuning approach based on the ANL module to enable fast task transfer on diffusion models with less tuning parameters.
> > >
> > > We appreciate for your valuable suggestions to improve our paper. We will submit a revised version based on them.

---

> > ### Author Response · Authors · 2022-12-11
> > **A kind reminder**
> >
> > Dear Reviewer #uRV2,
> >
> > Thanks again for all the time and effort in reviewing our paper. As you might know that the reviewer final recommendation deadline is approaching. We appreciate your valuable comments and have made detailed responses to them. We have also incorporated all your valuable suggestions to our revised manuscript.
> >
> > During the discussion phase between reviewers and AC, we added two experiments to show that (1) Our tuning approach shows better and more stable performance than the baseline with different learning rates. (2) Our tuning approach enables effective transfer learning while freezing the encoder (i.e., only tuning the block9-16) of pre-trained U-Net, which provides a insight to further reducing the tuned parameters.
> >
> > If our response has addressed your concerns, we would highly appreciate it if you could re-evaluate our work and consider raising the score. Also, we are happy to engage in discussions if you have any further questions on our work.
> >
> > Best regards,
> >
> > Authors of Paper #2648

---

### Official Review · Reviewer_2H8p · 2022-10-25

**Confidence:** 4
**Clarity, Quality, Novelty And Reproducibility:** 1. The paper is well-motivated by the…
**Correctness:** 4
**Technical Novelty And Significance:** 3
**Empirical Novelty And Significance:** 2
**Recommendation:** 6

**Strength And Weaknesses:**

Strengths

1. It proposes to transfer pretrained LDMs to small data and use limited training resources.

2. The CLIP embeddings are used as the conditions of LDMs and a portion of model parameters (Attention-NonLinear) are finetuned to be efficient.

3. Experiments demonstrate better FID scores than finetuning all the parameters and the GAN related methods.

Weaknesses

1. Some equation types exist. For example, the right of Equation 1 and the left of Equation 3 have one parenthesis missing, respectively. The Gaussian distribution in Equation 1 and 2 have inconsistent forms. Both d_c and d^c appear in the descriptions under Equation 8.

2. Table 2 uses the semi-supervised setting. It generates 10 additional samples for each labeled sample. Can we assume the 10 generated samples have the sample labels as their original sample? Since the generation is conditioned on the original sample, it’s very likely that the generated variants still keep the label. If so, it could be just supervised learning with data augmentations.

3. Section 3.3 mentions using masked images as conditional input. Are they used in training, generation, or both? If used in training, any ablation study of delta? Figure 6 only shows illustrations of masking used in generation.

4. I feel the task is defined clearly. The experiments use the LDM pre-trained on ImageNet. Is the pre-training task a noise-to-image generation task? In the finetuning, the task is an image-to-image generation task due to the image conditioning. This change is not described clearly in the paper.

5. Are there two factors of randomness in generating images, i.e., the random masking of the conditional image and the random Gaussian noises? I guess Figures 3 and 5 use random Gaussian noises but no image masking, and Figure 6 uses fixed Gaussian noises but random conditional image mask, right?

6. CLIP embedding and Attention-NonLinear are two components of the proposed transfer learning method. What if removing the CLIP embedding part, i.e., only finetuning the NonLinear part of the LDM? With CLIP, it is an image-to-image generation formulation since it generates one image conditioned on an image. It’s unclear for me how important the CLIP embedding is in the proposed method. Which one really matters in the low-data regime: the image-to-image generation formulation help (compared to the pure noise-to-image generation) or the CLIP embedding? Maybe also try a different embedder, e.g, a pre-trained vision transformer.


**Summary Of The Paper:**

This paper introduces an efficient method to finetune pre-trained LDM to small data. The problem is formulated as an image-to-image generation task. A conditional image is first encoded by the CLIP image encoder and then injected into the LDM by the cross attention mechanism. To avoid overfitting, only the parameters of attention and non-linear layers are finetuned. Experiments show the effectiveness both quantitatively and qualitatively.


**Summary Of The Review:**

This paper proposes an efficient fintuning method to adapt pre-trained LDM to small datasets. The method is well-motivated and the results are promising, but some details and ablations are missing. See the above comments.

---

> ### Author Response · Authors · 2022-11-11
> **Response to Reviewer #2H8p**
>
> We appreciate for acknowledging the motivation and clarity of our work. Here, we address your concerns as follows:
>
> **Q1:** Some equation typos exist.
>
> **A1:** Thanks for your careful reviews. (1) After double-checking our formulation, we guess you mean that the right of Equation 1 missed a parenthesis. We have revised the parenthesis typos in Equation 2 and 3. (2) The different form of Gaussian distribution in Equations 1 and 2 is correct. In the forward process, the Gaussian noise is defined artificially, while the Gaussian noise in reverse (generation) process is parameterized by a learnable network. (3) The typo in the descriptions under Equation 8 is revised.
>
> **Q2:** About the semi-supervised learning.
>
> **A2:** Yes. The generated samples have the same labels as the condition samples.
>
> **Q3:** About the masked image.
>
> **A3:** The mask mechanism is only adopted in generation.
>
> **Q4:** About the task definition of pre-training and fine-tuning.
>
> **A4:** We use the LDM pre-trained on ImageNet with classifier conditions. Generally speaking, both pre-trained and fine-tuned models belong to conditional generative models. The pre-trained model is guided by semantic labels, while the fine-tuned model is guided by semantic representations. Our tuning approach can be also applied to unconditional generation task (i.e., noise-to-image generation), please refer to Appendix A for more details.
>
> **Q5:** About the randomness in generation.
>
> **A5:** Yes. Only Figure 6 uses random conditional image mask. However, the Gaussian noise in generation stage is always random, the guidance is achieved by cross-attention mechanism.
>
> **Q6:** Ablation on CLIP embedding.
>
> **A6:** Thanks for your valuable suggestions. The mentioned ablation study is already provided in the original manuscript (see **Appendix A**). In Appendix A, we investigate unconditional generation by replacing the cross-attention module with the self-attention module. Of course, the CLIP embedding is not adopted in this setting. We can find the FID score of this variant is slightly lower than the conditional ones. It is worth noticing that our tuning approach (i.e., only fine-tuning the ANL parts) really matters in the low-data regime, the CLIP model is just used to further improve it. Our tuning approach can be also applied to unconditional generation, which is the so-called noise-to-image generation and removes the CLIP embedding (see Appendix A in our original manuscript). In our early experiments, we have also tried the ViT embedder motivated by [1], and found that the pre-trained ViT performs better than the learnable ViT (unfreeze ViT during fine-tuning), gets similar performance with CLIP. Considering that CLIP enables simple text-to-image generation, we use CLIP in our paper.
>
> **Q7:** About the reproducibility.
>
> **A7:** We will release our code upon publication.
>
> Thanks again for your valuable reviews. Please let us know if you have any further question on our work.
>
> Reference:
>
> [1] Giannone et al, Few-shot diffusion models. Arxiv’2022

---

> ### Author Response · Authors · 2022-12-12
> **A kind reminder**
>
> Dear Reviewer #2H8p,
>
> Thanks again for all the time and effort in reviewing our paper. As you might know that the reviewer final recommendation deadline is approaching. We appreciate your valuable comments and have made detailed responses to them. We have also incorporated all your valuable suggestions to our revised manuscript.
>
> During the discussion phase between reviewers and AC, we added two experiments to show that (1) Our tuning approach shows better and more stable performance than the baseline with different learning rates. (2) Our tuning approach enables effective transfer learning while freezing the encoder (i.e., only tuning the block9-16) of pre-trained U-Net, which provides a insight to further reducing the tuned parameters.
>
> If our response has addressed your concerns, we would highly appreciate it if you could re-evaluate our work and consider raising the score. Also, we are happy to engage in discussions if you have any further questions on our work.
>
> Best regards,
>
> Authors of Paper #2648

---

### Official Review · Reviewer_YaAB · 2022-10-26

**Confidence:** 4
**Correctness:** 4
**Technical Novelty And Significance:** 3
**Empirical Novelty And Significance:** 3
**Recommendation:** 6

**Clarity, Quality, Novelty And Reproducibility:**

The method is simple and clearly explained. It barely meets the technical novelty bar, but is very relevant to the work going on in the field today. The authors do not mention if they will release the code, so it is not clear how reproducible would be the method, but it does look simple enough for someone to try it out.

**Strength And Weaknesses:**

Strengths:
1) The method is simple to implement and shows strong results over good baselines.
2) It is all the more important given that new latent diffusion models like 'stable diffusion' are publicly available which are trained on hundreds of millions of images and fine-tuning on newer datasets with less data will become even more important. The paper proposes a neat trick to address this problem.

Weaknesses:
The cross attention based additional architecture isn't very novel and there have been several attempts around this framework traditional deep learning tasks.


**Summary Of The Paper:**

The paper proposes an efficient way of fine-tuning diffusion models on new datasets. Instead of training the entire network or training from scratch, the authors add a attention non-linear block which is the only learnable part which is learned during fine-tuning. The attention non-linear block is a cross attention layer followed by a 2 layer NN. Results are shown on a few datasets where the method shows better results.

**Summary Of The Review:**

The merits of the paper outweigh the weaknesses, so I am recommending the paper for acceptance. Would have given a 7 if there was an option, but its better than a 6, so I am marking it as an 8.

After reading reviews of other reviewers about concerns around novelty, I am updating my rating to 6.

---

> ### Author Response · Authors · 2022-11-11
> **Response to Reviewer #YaAB:**
>
> We appreciate for acknowledging the clarity and significance of our work. We will release our code upon publication. As for novelty of the cross-attention module, we want to emphasize the key contribution of our paper is the effective tuning approach based on this module, which enables a faster transfer learning for diffusion models. Thanks again for your valuable reviews. Please let us know if you have any further question on our work.

---

> > ### Comment · Reviewer_YaAB · 2022-11-27
> > **design space exploration**
> >
> > As mentioned by the AC, is not tuning the pre-trained LDM at all, always the best choice? May be it would be good to show a degradation in performance with different learning rates in one setting (like a dataset) for the pre-trained weights instead of a on/off kind of experiment currently being performed. Also, it would be good to see if we fine-tuned certain blocks and don't fine-tune other parts (like Prompt-to-Prompt Image Editing with Cross Attention Control), and how it affects performance in this low-shot setting. The design space exploration with these kind of experiments should strengthen the paper.

---

> > > ### Author Response · Authors · 2022-11-27
> > > **Thanks for your continued attention.**
> > >
> > > Thanks for the valuable suggestions given by the AC and reviewers. Based on our observations, our tuning approach performs better than the baseline TuningAll in transfer learning with few data and optimization steps. The choice of learning rate will affect the performance of tuning methods, and we will conduct an adequate experiment on it. The second suggestion is very interesting. If we understand correctly, [1] controls the attention injection by different diffusion steps rather than different blocks (figure 6 in [1]). The paper "Diffusion Models Already Have a Semantic Latent Space" [2] is closer to the mentioned insight, which provides a qualitative result on how does different block in diffusion models affects image manipulation (figure 15 in [2]). However, we want to emphasize that both [1] and [2] focus on image manipulation, but our paper focuses on general transfer learning with limited data and optimization steps, in which the domain gap between source and target images is large. In summary, we will conduct the following two experiments to strengthen our paper:
> > >
> > > 1. The performance of our method and the baseline TuningAll with different learning rates.
> > >
> > > 2. The ablation study on tuning certain blocks in low-shot transfer learning setting. This ablation will be conducted on our method (certain ANL modules) and the baseline TuningAll (certain residual blocks in U-Net).
> > >
> > > Thanks again for the valuable suggestions. We will upload the results of aforementioned experiments in a week.
> > >
> > > Reference:
> > >
> > > [1] Hertz et al,  Prompt-to-Prompt Image Editing with Cross Attention Control, Arxiv 2022.
> > >
> > > [2] Anonymous authors, Diffusion Models Already Have a Semantic Latent Space, ICLR 2023 submission.

---

> > > > ### Comment · Reviewer_YaAB · 2022-11-27
> > > > **reg [1,2]**
> > > >
> > > > yes, I understand that these methods focus on image manipulation. I was just mentioning that they fix weights till a certain layer and train the remaining ones, so similar kind of experiments can be tried here. Thanks for the correction, yes, [1] is about time steps and not blocks, [2] would be a better reference.

---

> > > > ### Author Response · Authors · 2022-12-01
> > > > **Please refer to the updated meta-response.**
> > > >
> > > > We have uploaded the results in the meta-response "Update: Two experiments are added.". Thanks for your continued attention and insightful suggestions.

---

### Author Response · Authors · 2022-11-16
**General Response: A revised manuscript is uploaded.**

We appreciate all the reviewers for their thorough reviews on our work. We have revised our manuscript based on their valuable suggestions. Here we provide a summary of the main changes:

1. We have revised the typos in equation 2 and 3, the unclear statement in contribution part.

2. In Section 3.3, we add a new paragraph to discuss unconditional generation.

3. In Table 1, we add the result of unconditional generation, which is originally presented in Appendix A.

4. In Section 4.2, we provide the ablation study on pre-trained encoders, and the comparison between conditional and unconditional models.

5. In Appendix A, we add the experimental result of different mask rate.

6. In Appendix B, we provide some insights on extending our method to other DPMs, and we are working on it.

In addition, we would like to kindly remind the reviewers that the deadline of rebuttal stage is approaching. Please let us know if you have any further question on our work. If the responses above are sufficient, we kindly ask you to consider raising the score accordingly. Thank you.

---

### Author Response · Authors · 2022-12-01
**Update: Two experiments are added.**

Based on the insightful suggestions from the AC and reviewers, we further add two experiments on (1) the sensitivity towards learning rates and (2) the chosen of fine-tuned blocks. All variants are evaluated on StandfordCars dataset, and 1k samples for each variant are generated for the FID calculation. The results and analyses are shown as follows:

**1.How does different learning rate affect transfer learning in diffusion models?**

In our default implementations, we scale up the learning rate by: base learning rate$\times$batch size $\times$the number of GPUs, and set the base learning rate as 2e-6. Firstly, we scale up/down the base learning rate by a factor of 10.

|Leanring Rate | 2e-8 | 2e-7 | 2e-6 | 2e-5 | 2e-4 |
| :-----| :-----: | :-----: | :-----: | :-----: | :-----: |
|TuningAll (1500steps) | 68.89 | 33.13 | 33.47 | N/A(352.00) | N/A |
|TuningAll (3000steps) | 42.84|32.07|28.03|N/A(370.04)|N/A|
|Ours (1500steps)| 51.30| 30.15|26.66|53.00|N/A|
|Ours (3000steps)|36.97|28.33|24.62|33.50|N/A|

From this table, we can observe that a low learning rate is preferred since the weights of U-Net are pre-trained. Compared to the baseline TuningAll, our method always gets better performance in each setting, and is more robust w.r.t. learning rates. For examples, TuningAll fails with the 2e-5 learning rate, while our method achieves 33.5 FID with 3k steps. In practice, the learning rate from 2e-7 to 2e-6 is suggested:

|Learning Rate | 2e-7 | 4e-7 | 6e-7 | 8e-7 | 1e-6 | 2e-6|
| :-----| :-----: | :-----: | :-----: | :-----: | :-----: | :-----: |
|TuningAll (1500steps) | 31.13| 37.05| 36.19|35.16|38.42|33.47|
|TuningAll (3000steps) | 32.07|31.42|33.53|34.38|35.62|28.03|
|Ours (1500steps)| 30.15|29.33|29.07|28.73|28.78|26.66|
|Ours (3000steps)|26.33|25.81|24.96|24.02|24.83|24.62|

**2.The ablation study on tuning certain blocks in low-shot setting.**

In detailed implementations, there are totally 16 blocks (of course, 16 ANL modules) in U-Net. Following the insight from [1], we conduct ablation study on tuning certain blocks (ANL modules). Specifically, we validate three variants: tuning blocks5-16, blocks9-16 and blocks13-16.

|Tuning Blocks | all blocks |blocks5-16 | blocks9-16 | blocks13-16 |
| :-----| :-----: | :-----: | :-----: | :-----: |
|TuningAll (1000steps) | 40.27|83.23|145.65|238.73|
|TuningAll (3000steps) | 28.03|61.00|109.87|175.14|
|TuningAll (3000steps) | 27.49|69.15|81.32|97.38|
|Ours (1000steps)|29.98 |30.97|41.49|145.70|
|Ours (3000steps)| 24.62|25.70|32.12|109.90|
|Ours (5000steps)| 20.33|22.08|24.47|81.28|


We can observe that (1) Tuning certain blocks is not suitable for TuningAll since it suffers from a clear performance degradation with even freezing blocks1-4. (2) Freezing the ANL modules in the encoder (i.e., tuning blocks5-16 or blocks9-16) has limited impact on our tuning method. For instance, our tuning method (5k steps) gains 22.08 and 24.47 FID with tuning blocks5-16 and blocks9-16, respectively, which is close to tuning all ANL modules. (3) For transferring pre-trained diffusion models, the decoder is more important than the encoder. Even freezing the encoder part, our tuning approach still achieves competitive performance. To further validate this claim, we conduct this ablation study in a reverse version, and find that tuning blocks1-12 or blocks1-8 is much worse than its symmetrical variant (tuning blocks5-16 or blocks9-16). This ablation study provides a convincing direction for further reducing tuning parameters.

Thanks again for the insightful reviews and suggestions from the AC and reviewers. We are happy to engage in discussions if you have any further question on our work.

Reference:

[1] Anonymous authors, Diffusion Models Already Have a Semantic Latent Space, ICLR 2023 submission.

---

### Decision · Program_Chairs · 2023-01-20

**Decision:**

Reject

**Justification For Why Not Higher Score:**

Reviewers share some major concerns about this paper. In particular, they thought fine-tuning a subset of parameters is nothing new and has already been a technique that is widely known in the field. Also, the proposed algorithm is heavily relied on the LDM architecture, which makes it unclear about the generalization ability on standard diffusion models or other models, rather than only LDM. At last, though there might lack relevant comparison works in the literature, the authors only designed quite weak baselines for the comparison experiments.


**Justification For Why Not Lower Score:**

N/A

**Metareview: Summary, Strengths And Weaknesses:**

This paper introduced a method to fine tune large generative models on small datasets. In particular, the authors used CLIP to encode images from the small datasets into a semantic latent space, and frozen pre-trained model via a cross-attention module. The authors have done basic experiments to evaluate the performance of the proposed algorithm.